# Acting without considering personal costs signals trustworthiness in helpers but not punishers
Nicole C. Engeler[1] ✉ & Nichola J. Raihani[1,2]

Third-party punishment and helping can signal trustworthiness, but the interpretation of deliberation may vary: uncalculated help signals trustworthiness, but this may not hold for punishment. Using online experiments, we measured how deliberation over personal costs and impacts to targets affected the trustworthiness of helpers and punishers. We expected that personal cost-checking punishers and helpers would be trusted less. Conversely, impact deliberation was expected to increase the perceived trustworthiness of punishers but not helpers. Replicating previous work, we found that refraining from checking the personal cost of helping signals trustworthiness (although evidence for observers trusting uncalculating over calculating helpers was mixed). This did not extend to punishment: only uncalculating *non*-punishers were more trustworthy than cost-checking non-punishers. Impact deliberation results were mixed: deliberation affected the trust and trustworthiness of non-helpers more than helpers and no conclusive results were found for punishment. These results show that deliberation differentially affects assessments of those who help or punish others.

**Protocol registration**

The Stage 1 protocol for this Registered Report was accepted in principle on 13th November 2023. The protocol, as accepted by the journal, can be found at https://doi.org/10.6084/m9.figshare.24559462.v1.

Prosocial behaviours, such as helping and cooperating, can benefit others but often come at a personal cost to the actor[1–3]. Punishment, which involves an actor paying a cost to impose a cost on a social partner[4,5], can encourage and maintain prosocial behaviours by deterring selfish actions[6–13]. Although punishing anti-social behaviours can increase group-level cooperation, it also imposes a cost on the punisher by requiring effort and time, and puts the punisher at risk of retaliation[6,14,15]. To understand why people invest in punitive acts, we must explain how punishment might ultimately lead to downstream benefits for the punisher.

This question is particularly pertinent when it comes to third-party punishment, where a punisher intervenes to punish a cheat even though they were not personally harmed by the cheat's behaviour and may not interact with the target of punishment again in the future. Third-party punishment can still provide reputation benefits to the punisher[16–21], either by signalling their formidability (which may deter their current social partners or bystanders from transgressing in the future[18,22]) or by signalling

their cooperative intent (which may result in others being more likely to cooperate with them[23] or choose them as partners for cooperative interactions[16,24–27]). Third-party punishment can therefore act as a signal that communicates an otherwise unobservable intent to act prosocially[21,23,26–30]. Accordingly, in some settings, individuals invest more in third-party punishment when they are observed[20,24] and are evaluated in a preferential manner by others for doing so[26].

Nevertheless, punishment is, by definition, a harmful act, which complicates inferences about the punisher's intentions[5,21,24]. Punishment could stem from antisocial, competitive, or spiteful motivations rather than from a desire to cooperate, promote fairness, or uphold social norms[5]. Indeed, compared to those who compensate victims, third-party punishers typically have higher scores for antisocial personality traits such as Machiavellianism, narcissism, and psychopathy[31]. Third-party punishment is therefore a more ambiguous signal of trustworthiness and cooperative intent than helping or compensating a victim[21,24,26,31,32]; and is most likely to

[1]Experimental Psychology, University College London, London, UK. [2]School of Psychology, University of Auckland, Auckland, New Zealand.
✉e-mail: engeler.nicole@gmail.com

signal cooperative intent in scenarios where the punisher cannot compensate the victim, and where self-serving motives are less likely[21] (e.g. when the punisher does not increase their own payoffs relative to those of the target when they punish[5,21]).

The potential for helpful acts to signal cooperative intent also depends, to some extent, on context[33]. One such context concerns whether the helpful act was calculated or not: uncalculated help is a stronger signal of the helper's cooperative disposition than calculated help. One recent study operationalised uncalculated help by measuring whether individuals looked at the personal cost to themselves before helping, and by recording how long it took individuals to make their helping decision once the cost of helping was revealed[26]. Helpers who check the cost are ostensibly weighing up the costs and benefits of their actions, suggesting that cooperating is a more strategic and calculated move. Response time is also informative about a person's underlying commitment to cooperation, as slower decisions indicate greater decision conflict[34]. Observers use information about another's decision time to infer levels of conflict experienced and to make predictions about whether a person is making a calculated or uncalculated decision to cooperate[26,35–37], and whether to trust them[26]. This previous work found that uncalculated cooperation was a more reliable signal of trustworthiness than calculated cooperation[26]. Moreover, people were apparently aware of the signalling value of uncalculated cooperation and were less likely to check the costs of cooperating and made cooperative decisions more quickly when observed[26].

Although both third-party punishment and helping are prosocial acts, decision conflict for these behaviours may not be interpreted in the same way, and it is therefore unclear whether findings on uncalculated help[26] would be expected to translate directly to the punishment setting. We address this issue here. Decision conflict over whether to help another is likely to stem from self-interested considerations of whether to pay a personal cost. Decision conflict over punishment, by contrast, could also stem from concerns about inflicting harm on the target. This perspective yields nuanced predictions about the two different measures of decision conflict above. Checking the personal cost of administering punishment is likely to indicate a self-interested concern about personal costs. As with helping, such calculated decisions may be perceived negatively by observers. However, because punishment involves imposing costs on another individual, taking longer to decide whether to punish another person might not be viewed negatively and could even be viewed positively. Perhaps carefully thinking about and balancing both the prosocial aspect as well as the negative consequences for the punished is the 'right' thing to do when deciding whether to engage in third-party punishment.

To better define the conditions under which punishers are viewed positively and to differentiate between punishment and helping as signals of trustworthiness, we conducted two studies. Study 1 aimed to replicate Jordan et al.'s[26] research on uncalculated helping and extend it to punishment, asking how deliberation over personal costs affects trustworthiness perceptions in both cases. Study 2 extended this by asking how deliberation over the impacts on targets affects perceptions of trustworthiness for both punishment and helping. With this approach we hoped to understand whether and why punishers are evaluated differently to helpers, and to show how deliberation differentially signals trustworthiness in decisions to punish or help others. Overall, we expected to show that deliberating about the *personal cost* of one's actions signals untrustworthiness for both punishing and helping, whereas deliberating about the *impact* of those actions on others constitutes the differentiating factor between helping and punishing behaviours. Specifically, we expected that those who deliberate about the impact of punishment are viewed relatively positively, whereas those who deliberate about the impact of helping are viewed relatively negatively. See Table 1 for descriptions of all preregistered hypotheses.

Across two studies, comprising five experiments (Table 2), we investigated whether and when uncalculated punishment and help are used as signals of trustworthiness. As in Jordan et al.[26], all experiments had two stages: a first stage, where Player A could pay a cost to help a victim/punish a cheat; and a second stage, where Player B decided whether and how much to trust Player A. Any money entrusted by Player B was tripled by the

experimenter and Player A then decided how much to return to Player B, yielding a measure of trustworthiness. We included two conditions: Player B was either be able to make their trusting decision based on (i) Player A's decision and decision process in the first stage (via cost-checking/decision time), or (ii) solely on Player A's help/punishment decision, with the decision *process* remaining concealed.

Four of our five experiments operationalised deliberation through cost- or impact-checking behaviours. Due to financial constraints, we included only one decision time study, specifically focussing on personal costs of punishment. This context holds particular significance, as we expected to observe significant differences between decision time and cost checking. Longer decision times may be attributed to concern for the target, in addition to self-interested considerations. Because decision conflict over punishment could also stem from concerns about inflicting harm on the target, this experiment forms a bridge between Studies 1 (personal cost deliberation) and 2 (target impact deliberation).

Study 1 investigated whether and how personal cost deliberation signals trustworthiness. Calculated behaviours were operationalised as decisions to check the personal cost of helping/punishing (1.1 & 1.2a), or long decision times after the cost of punishing is revealed (1.2b). Study 1 therefore comprises three separate experiments (Table 2), investigating how deliberating over decisions to help (1.1) or to punish (1.2a & 1.2b) signal trustworthiness.

Study 2 was similar to Study 1 but investigated whether deliberation over impacts on targets of help/punishment signals trustworthiness. In Study 2, the personal cost of help/punishment was therefore known. Calculated behaviours were instead defined as decisions to check the impact of help/punishment ('impact checking', 2.1 & 2.2).

When deliberation was operationalised as checking the personal cost of helping/punishing (Experiments 1.1 and 1.2), we predicted that uncalculated decisions signal trustworthiness: individuals who do not check the cost of helping/punishing would be entrusted with more money than those who do check the cost. We also expected participants to be sensitive to these potential reputation benefits and to be less likely to check the cost of helping/punishing when they were observed than when their decision process was hidden from Player B. Further, we expected that individuals who do not check the cost of helping/punishing would be more trustworthy than those who do check the cost. Lastly, we anticipated uncalculated help to be a stronger signal of trustworthiness than uncalculated punishment.

When personal cost deliberation was operationalised as decision speed (Experiment 1.2b), our predictions were more nuanced. Here, the cost of punishing was revealed to Player A right before they made their decision. As the sole new information provided to participants prior to measuring their decision-making time related to the personal cost of punishing, we predicted that here too uncalculated punishment would be used as a signal of trustworthiness. Specifically, we expected that participants would exhibit faster decision-making when their decision time was revealed to others, in contrast to when the decision process was concealed and unable to confer any reputation-related advantages, as was observed with helping in Jordan et al.[26].

Nevertheless, we also envisaged some differences in how quick decisions to punish might be perceived by others, which would be driven by the different motives attributed to punishers and helpers. Jordan et al.[26] found that fast decisions to help others were perceived positively, likely because fast decisions were associated with less decision conflict. However, because punishment inflicts harm on the target, fast decisions to punish could be evaluated differently[5,21]. One possibility is that observers interpret a fast decision to punish as the punisher being moral and interested in restoring fairness regardless of the cost to the self. Here, uncalculated punishment would be approved of, and observers would infer that fast punishers were more trustworthy than slow punishers. Another possibility, however, is that observers may approve of more considered decisions to punish others if they infer that decision conflict stems from concern about the harm caused to the target. Thus, slow punishers may not be evaluated as negatively and, consequently, we expected decision speed to be a weaker signal of trustworthiness than cost-checking decisions for punishment.

## Table 1 | Design Table

| Question Primary Hypotheses | Hypothesis | Sampling plan (e.g. power analysis) | Analysis Plan | Interpretation given to different outcomes |
|---|---|---|---|---|
| Q1. Are uncalculated decisions around the personal cost of helping/punishing used as a signal of trustworthiness? | Helping: H1.1) Participants will be significantly less likely to check the cost of helping in the decision process observable condition than in the decision process hidden condition. Punishing: H1.2a) Participants will be significantly less likely to check the cost of punishment in the decision process observable condition than in the decision process hidden condition. H1.2b) Participants will make significantly faster punishing decisions in the decision process observable condition than in decision the process hidden condition. | Please refer to the Sampling plan in Methods for detail. H1.1 & H1.2a) The sample size for this model will be $N = 1306$ (653 Players A per condition in Experiment 1/2). H1.2b) The sample size for this model will be $N = 1306$ (653 Players A per condition in Experiment 3). | H1.1 & H1.2a) We will run a logistic regression with checking decision (0 = did not check the cost, 1 = checked the cost) as a function of decision process observability (0 = process hidden, 1 = process observable). H1.2b) We will run a linear regression, predicting decision time as a function of decision process observability (0 = process hidden, 1 = process observable). If the amount of time spent deciding whether to punish is highly skewed, punishing decision time will be natural log transformed. | H1.1 & H1.2a) A significant negative coefficient for observability (0 = decision process hidden, 1 = decision process observable) will be interpreted as evidence that participants are less likely to check the personal cost of helping/punishing when their decision process is observable compared to hidden (and therefore, that they are more likely to act uncalculatingly when their decision process can be observed). Otherwise, there is no evidence for H1.1/H1.2a. H1.2b) A significant negative coefficient for observability (0 = decision process hidden, 1 = decision process observable) will be interpreted as evidence that participants make faster punishing decisions (i.e., act uncalculatingly) when their decision process is observable compared to hidden. Otherwise, there is no evidence for H1.2b. |
| Q2. Are uncalculated decisions around the personal cost of helping/punishing perceived as a signal of trustworthiness? | Helping; H2.1) Observers will send significantly more of their endowment to helpers who did not check the personal cost of helping than to helpers who checked the cost. Punishing: H2.2a) Observers will send significantly more of their endowment to punishers who did not check the personal cost of punishing than to punishers who checked the cost. H2.2b) Observers will send significantly more of their endowment to punishers who made relatively fast (vs relatively slow) decisions to punish. | Please refer to the Sampling plan in Methods for detail. H2.1 & H2.2a) The sample size for this model will be $N = 653$ (Players B in the observable condition in Experiment 1/2). H2.2b) The sample size for this model will be $N = 653$ (Players B in the observable condition in Experiment 3). | Analyses will be restricted to the observable condition because Players B can only condition their trust on Player A decision processes in this condition. As Players B make two sending decisions (based on the two possible decisions made by Player A during the first stage), each sending decision will be treated as an observation and robust SEs will be clustered on observer ID to account for the non-independence of repeated observations from the same participant. The endowment sent will be transformed from pence sent to percentage of endowment sent for ease of interpretation. H2.1 & H2.2a) We will run a linear regression predicting the percentage of endowment sent by Players B as a function of helpers/punishers cost-checking decisions (0 = did not check the personal cost, 1 = checked the cost). H2.2b) We will run a linear regression predicting the percentage of endowment Players B sent to punishers in the observable condition as a function of decision time (0 = relatively slow, 1 = relatively fast). | H2.1 & H2.2a) A significant negative coefficient for cost checking (0 = did not check the cost, 1 = checked the cost) will be interpreted as evidence that observers send a higher proportion of their endowment to helpers/punishers who did not check the cost of helping/punishing compared to helpers/punishers who checked the cost of helping/punishing (i.e., that observers trust uncalculating helpers/punishers more than calculating helpers/punishers). Otherwise, there is no evidence for H2.1/H2.2a. H2.2b) A significant positive coefficient of decision speed (0 = relatively slow, 1 = relatively fast) will be interpreted as evidence that observers send a higher proportion of their endowment to punishers who decided relatively quickly compared to punishers who decided relatively slowly (i.e., that observers trust uncalculating punishers more than calculating punishers). Otherwise, there is no evidence for H2.2b. |
| Q3. Does the operationalisation of uncalculating behaviour differentially influence the perceived trustworthiness of punishers in the context of personal cost? | H3) Observers will send significantly less of their endowment to punishers who check the personal cost of punishing than to punishers who take a long time to decide to punish. | Please refer to the Sampling plan in Methods for detail. H3) The sample size for this model will be $N = 1306$ (Players B in the observable condition in Experiment 2 and 3). | H3) We will run a linear regression predicting the percentage of endowment sent by Players B as a function of experiment (decision time vs cost checking) and deliberation (uncalculated vs calculated), as well as the interaction between experiment and deliberation. | H3) A significant negative coefficient for the interaction between experiment (0 = decision time, 1 = cost checking) and deliberation (0 = uncalculated, 1 = calculated) will be interpreted as evidence that the effect of calculated vs uncalculated punishment on trust is stronger for cost checking than decision time. Otherwise, there is no evidence for H3. |
| Q4. Do uncalculated helping and punishing decisions differentially influence perceived trustworthiness in the context of personal cost? | H4) Observers will send significantly less of their endowment to helpers who check the personal cost than to punishers who check the personal cost. | Please refer to the Sampling plan in Methods for detail. H4) The sample size for this model will be $N = 1306$ (Players B in the observable condition in Experiment 1 and 2). | H4) We will run a linear regression predicting the percentage of endowment sent by Players B as a function of behaviour (punishing vs helping) and deliberation (uncalculated vs calculated), as well as the interaction between behaviour and deliberation. | H4) A significant negative coefficient for the interaction between behaviour (0 = punishing, 1 = helping) and deliberation (0 = uncalculated, 1 = calculated) will be interpreted as evidence that the effect of calculated vs uncalculated decisions on trust is stronger for helping than punishing. Otherwise, there is no evidence for H4. |
| Q5. Are uncalculated decisions around target impact used as a signal of trustworthiness? | Helping: H5.1) Participants will be significantly less likely to check the impact of helping in the decision process observable condition than in the decision process hidden condition. Punishing: H5.2) Participants will be significantly more likely to check the impact of punishment in the decision process observable condition than in the decision process hidden condition. | Please refer to the Sampling plan in Methods for detail. H5.1 & H5.2) The sample size for this model will be $N = 1306$ (653 Players A per condition in Experiment 4/5). | H5.1 & H5.2) We will run a logistic regression with checking decision (0 = did not check the impact, 1 = checked the impact) as a function of decision process observability (0 = process hidden, 1 = process observable). | H5.1) A significant negative coefficient for observability (0 = decision process hidden, 1 = decision process observable) will be interpreted as evidence that participants are less likely to check the impact of helping when their decision process is observable compared to hidden (and therefore, that they are more likely to act uncalculatingly when their decision process can be observed). Otherwise, there is no evidence for H5.1. H5.2) A significant positive coefficient for observability (0 = decision process hidden, 1 = decision process observable) will be interpreted as evidence that participants are more likely to check the impact of punishing when their decision process is observable compared to hidden (and therefore, that they are more likely to act uncalculatingly when their decision process can be observed). Otherwise, there is no evidence for H5.2. |

## Table 1 (continued) | Design Table

| Question Primary Hypotheses | Hypothesis | Sampling plan (e.g. power analysis) | Analysis Plan | Interpretation given to different outcomes |
|---|---|---|---|---|
| Q6. Are uncalculated decisions around target impact perceived as a signal of trustworthiness? | Helping: H6.1) Observers will send significantly more of their endowment to helpers who did not check the impact of helping on targets than to helpers who checked the impact. Punishing: H6.2) Observers will send significantly more of their endowment to punishers who checked the impact of punishing on targets than to punishers who did not check the impact. | Please refer to the Sampling plan in Methods for detail. H6.1 & H6.2) The sample size for this model will be $N = 653$ (Players B in the observable condition in Experiment 4/5). | Analyses will be restricted to the observable condition because Players B can only condition their trust on Player A decision processes in this condition. As Players B make two sending decisions (based on the two possible decisions made by Player A during the first stage), each sending decision will be treated as an observation and robust SEs will be clustered on observer ID to account for the non-independence of repeated observations from the same participant. The endowment sent will be transformed from pence sent to percentage of endowment sent for ease of interpretation. H6.1 & H6.2) We will run a linear regression predicting the percentage of endowment sent by Players B as a function of helpers/punishers impact-checking decisions (0 = did not check the impact, 1 = checked the impact). | H6.1) A significant negative coefficient for impact checking (0 = did not check the impact, 1 = checked the impact) will be interpreted as evidence that observers send a higher proportion of their endowment to helpers who did not check the impact of helping compared to helpers who checked the impact of helping (i.e., that observers trust uncalculating helpers more than calculating helpers). Otherwise, there is no evidence for H6.1. H6.2) A significant positive coefficient for impact checking (0 = did not check the impact, 1 = checked the impact) will be interpreted as evidence that observers send a higher proportion of their endowment to punishers who checked the impact of punishing compared to punishers who did not check the impact of punishing (i.e., that observers trust calculating punishers more than uncalculating punishers). Otherwise, there is no evidence for H6.2. |
| **Secondary Hypotheses** | | | | |
| Q7. Do uncalculated decisions around personal cost affect the perceived trustworthiness of non-helpers/non-punishers? | Helping: H7.1) Observers will send significantly more of their endowment to non-helpers who did not check the cost of helping than to non-helpers who checked the cost of helping. Punishing: H7.2a) Observers will send significantly more of their endowment to non-punishers who did not check the cost of punishing than to non-punishers who checked the cost of punishing. H7.2b) Observers will send significantly more of their endowment to relatively fast non-punishers than to relatively slow non-punishers. | Please refer to the Sampling plan in Methods for detail. H7.1 & H7.2a) The sample size for this model will be $N = 653$ (Players B in the observable condition in Experiment 1/2). H7.2b) The sample size for this model will be $N = 653$ (Players B in the observable condition in Experiment 3). | Analysis will be restricted to the observable condition because Players B can only condition their trust on Player A decision processes in this condition. As each observer makes two sending decisions, we will cluster robust SEs on observer ID. H7.1 & H7.2a) We will run a linear regression predicting the percentage of endowment sent by Players B to non-helpers/non-punishers as a function of cost checking decisions. H7.2b) We will run a linear regression predicting the percentage of endowment Players B sent to non-punishers as a function of decision speed. | H7.1) A significant positive coefficient for cost checking (0 = did not check the cost, 1 = checked the cost) will be interpreted as evidence that observers send more of their endowment to non-helpers who did not check the cost of helping compared to non-helpers who did not check the cost of helping (i.e., observers trust calculating non-helpers). Otherwise, there is no evidence for H7.1. H7.2a) A significant negative coefficient for cost checking (0 = did not check the cost, 1 = checked the cost) will be interpreted as evidence that observers send more of their endowment to non-punishers who did not check the cost of punishing compared to non-punishers who checked the cost (i.e., observers trust uncalculating non-punishers more than calculating non-punishers). Otherwise, there is no evidence for H7.2a. H7.2b) A significant positive coefficient for decision speed (0 = relatively slow, 1 = relatively fast) will be interpreted as evidence that observers send more of their endowment to non-punishers who take little time in their decision not to punish compared to those who take a long time to decide not to punish (i.e., observers trust uncalculating non-punishers more than calculating non-punishers). Otherwise, there is no evidence for H7.2b. |
| Q8. Do uncalculated decisions around personal cost have a stronger effect on the perceived trustworthiness of helpers/punishers than non-helpers/non-punishers? | Helping: H8.1) Players B will send significantly less of their endowment to helpers who checked the cost of helping than to non-helpers who checked the cost of helping. Punishing: H8.2a) Players B will send significantly less of their endowment to punishers who checked the cost of punishing than to non-punishers who checked the cost of punishing. H8.2b) Players B will send significantly less of their endowment to relatively slow punishers than to relatively slow non-punishers. | Please refer to the Sampling plan in Methods for detail. H8.1 & H8.2a) The sample size for this model will be $N = 653$ (Players B in the observable condition in Experiment 1/2). H8.2b) The sample size for this model will be $N = 653$ (Players B in the observable condition in Experiment 3). | H8.1 & H8.2a) We will run a linear regression predicting the percentage of endowment sent by Players B as a function of helping/punishing decision, cost checking decision, and the interaction between the two. Robust SEs will be clustered on observer ID, as Players B will make four sending decisions based on each of the four possible Player A choices. H8.2b) We will run a linear regression predicting the percentage of endowment sent by Players B as a function of punishing decision, decision speed and the interaction between the two. Robust SEs will be clustered on participant ID, accounting for repeated observations (four per participant). | H8.1 & H8.2a) A significant negative coefficient for the interaction between helping/punishing (0 = did not help/punish, 1 = helped/punished) and cost checking decisions (0 = did not check the cost, 1 = checked the cost) will be interpreted as evidence that the effect of uncalculating behaviour on trust is larger when Players A decided to help/punish compared to when Players A decided not to help/punish. Otherwise, there is no evidence for H8.1/H8.2a. H8.2b) A significant negative interaction between punishing decision (0 = did not punish, 1 = punished) and decision speed (0 = relatively slow, 1 = relatively fast) will be interpreted as evidence that the effect of uncalculating behaviour on trust is larger when Player As decided to punish compared to when Player As decided not to punish. Otherwise, there is no evidence for H8.2b. |

**Table 1 (continued) | Design Table**

| Question Primary Hypotheses | Hypothesis | Sampling plan (e.g. power analysis) | Analysis Plan | Interpretation given to different outcomes |
|---|---|---|---|---|
| 9. Do uncalculated decisions around personal cost have a stronger effect on the trustworthiness of helpers/ punishers than non-helpers/non-punishers? | Helping:H9.1) Helpers who checked the cost of helping will return significantly less of their endowment than non-helpers who checked the cost of helping. Punishing:H9.2a) Punishers who checked the cost of punishing will return significantly less of their endowment than non-punishers who checked the cost of punishing. H9.2b) Fast punishers will return significantly less of their endowment than fast non-punishers. | Please refer to the Sampling plan in Methods for detail. H9.1 & H9.2a) The sample size for this model will be $N = 1306$ (653 Players A per condition in Experiment 1/2). H9.2b) The sample size for this model will be $N = 1306$ (653 Players A per condition in Experiment 3). | H9.1 & H9.2a) We will run a linear regression predicting the percentage of endowment returned by Players A as a function of helping/punishing decision, cost checking decision, as well as the interaction between the two. H9.2b) We will run a linear regression predicting the percentage of endowment returned by Players A as a function of punishing decision, log-transformed punishing decision time, their interaction, as well as log-transformed general comprehension speed. As the analysis is correlational, we wish to avoid concerns that the punishing decision time is reflective of general comprehension and reading speed rather than only of the time taken to consider whether to punish. Therefore, the natural log-transformed time spent reading the comprehension questions (i.e., the sum of time spent on the three comprehension question pages) will be included as a control for comprehension and reading speed. | H9.1 & H9.2a) A significant negative coefficient for the interaction between helping/punishing (0 = did not help/punish, 1 = helped/punished) and cost checking decisions (0 = did not check the cost, 1 = checked the cost) will be interpreted as evidence that the effect of uncalculating decision making on trustworthiness is larger when Players A decide to help/punish compared to when Players A decide not to help/punish. Otherwise, there is no evidence for H9.1/H9.2a. H9.2b) A significant negative interaction between punishing decision (0 = did not punish, 1 = punished) and log-transformed decision time will be interpreted as evidence that decision time is a stronger predictor of untrustworthiness when Player A punished versus did not punish. Otherwise, there is evidence for H9.2b. |
| Q10. Does the operationalisation of uncalculating behaviour differentially influence the perceived trustworthiness of non-punishers in the context of personal cost? | H10) Observers will send significantly less of their endowment to non-punishers who check the personal cost of punishing than non-punishers who take a long time to decide. | Please refer to the Sampling plan in Methods for detail. H10) The sample size for this model will be $N = 1306$ (Players B in the observable condition in Experiment 2 and Experiment 3). | H10) We will run a linear regression predicting the percentage of endowment sent by Players B as a function of experiment (decision time vs cost checking) and deliberation (uncalculated vs calculated), as well as the interaction between experiment and deliberation. | H10) A significant negative coefficient for the interaction between experiment (0 = decision time, 1 = cost checking) and deliberation (0 = uncalculated non-punishment, 1 = calculated non-punishment) will be interpreted as evidence that the effect of calculated vs uncalculated non-punishment on trust is stronger for cost checking than decision time. Otherwise, there is no evidence for H10. |
| Q11. Do uncalculated decisions around target impact affect the perceived trustworthiness of non-helpers/non-punishers? | Helping:H11.1) Observers will send significantly more of their endowment to non-helpers who checked the impact of helping than to non-helpers who did not check the impact. Punishing:H11.2) Observers will send significantly more of their endowment to non-punishers who checked the impact of punishing than to non-punishers who did not check the impact. | Please refer to the Sampling plan in Methods for detail. H11.1 & H11.2) The sample size for this model will be $N = 653$ (Players B in the observable condition in Experiment 4/5). | Analysis will be restricted to the observable condition because Players B can only condition their trust on Player A decision processes in this condition. As each observer makes two sending decisions, we will cluster robust SEs on observer ID. H11.1 & H11.2) We will run a linear regression predicting the percentage of endowment Player Bs sent to non-helpers/ non-punishers as a function of impact checking decisions. | H11.1 & H11.2) A significant positive coefficient for impact checking (0 = did not check the impact, 1 = checked the impact) will be interpreted as evidence that observers send more of their endowment to non-helpers/non-punishers who checked the impact of helping/punishing compared to non-helpers/non-punishers who did not check the impact of helping/punishing (i.e., observers trust calculating non-helpers/non-punishers more than uncalculating non-helpers/non-punishers). Otherwise, there is no evidence for H11.1/H11.2. |
| Q12. Do uncalculated decisions around target impact have a stronger effect on the perceived trustworthiness of helpers/ punishers than non-helpers/non-punishers? | Helping:H12.1) Observers will send significantly less of their endowment to helpers who checked the impact of helping than to non-helpers who checked the impact of helping. Punishing:H12.2) Observers will send significantly less of their endowment to punishers who checked the impact of punishing than to non-punishers who checked the impact of punishing. | Please refer to the Sampling plan in Methods for detail. H12.1 & H12.2) The sample size for this model will be $N = 653$ (Players B in the observable condition in Experiment 4/5). | H12.1 & H12.2) We will run a linear regression predicting the percentage of endowment sent by Players B as a function of helping/punishing decision, impact checking decision, and the interaction between the two. Robust SEs will be clustered on observer ID, as Players B will make sending decision based on each of the four possible Player A choices. | H12.1 & H12.2) A significant negative coefficient for the interaction between helping/punishing (0 = did not help/punish, 1 = helped/punished) and impact checking decisions (0 = did not check the impact, 1 = checked the impact) will be interpreted as evidence that the effect of uncalculating behaviour on trust is larger when Players A decided to help/punish compared to when Players A decided not to help/punish. Otherwise, there is no evidence for H12.1/H12.2. |
| Q13. Do uncalculated decisions around target impact have a stronger effect on the actual trustworthiness of helpers/ punishers than non-helpers/non-punishers? | Helping:H13.1) Helpers who checked the impact of helping will return significantly less of their endowment than non-helpers who checked the impact of helping. Punishing:H13.2) Punishers who checked the impact of punishing will return significantly less of their endowment than non-punishers who checked the impact of punishing. | Please refer to the Sampling plan in Methods for detail. H13.1 & H13.2) The sample size for this model will be $N = 1306$ (653 Players A per condition in Experiment 4/5). | H13.1 & H13.2) We will run a linear regression predicting the percentage of endowment returned by Players A as a function of helping/punishing decision, impact checking decision, as well as the interaction between the two. | H13.1 & H13.2) A significant negative coefficient for the interaction between helping/punishing (0 = did not help/punish, 1 = helped/punished) and impact checking decisions (0 = did not check the impact, 1 = checked the impact) will be interpreted as evidence that the effect of uncalculating decision making on trustworthiness is larger when Players A decide to help/punish compared to when Players A decide not to help/punish. Otherwise, there is no evidence for H13.1/H13.2. |

**Table 1 (continued) | Design Table**

| Question / Primary Hypotheses | Hypothesis | Sampling plan (e.g. power analysis) | Analysis Plan | Interpretation given to different outcomes |
|---|---|---|---|---|
| **Preregistered Exploratory Hypotheses** For all exploratory hypotheses, if power requirements are not achieved, the results will be reported as suggestive, pending confirmation in future research. | | | | |
| Q14. Do uncalculated decisions around the personal cost of helping/punishing predict trustworthiness? | Helping: H14.1) Helpers who did not check the cost of helping will return significantly more of their endowment than helpers who checked the cost of helping. Punishing: H14.2a) Punishers who did not check the cost of punishing will return significantly more of their endowment than punishers who checked the cost. H14.2b) Punishers who made faster decisions to punish will return significantly more of their endowment than punishers who took a longer time to decide to punish. | Please refer to the Sampling plan in Methods for detail. H14.1 & H14.2a) As we do not know how many Player As will decide to help/punish, the sample size for this model will be up to N = 1306 (653 Player As per condition in Experiment 1/2). H14.2b) As we do not know how many Player As will decide not to punish, the sample size for this model will be up to N = 1306 (653 Players A per condition in Experiment 3). | Here, both the observable and the hidden condition will be used, as we collect the data on Player A's decision process, even when Player B cannot observe it. H14.1 & H14.2a) We will run a linear regression predicting the percentage returned as a function of the helpers/ punishers cost checking decision. H14.2b) We will run a linear regression predicting the percentage of endowment returned by punishers as a function of log-transformed decision time and log-transformed general comprehension speed. | H14.1 & H14.2a) A significant negative coefficient of cost checking (0 = did not check the cost, 1 = checked the cost) will be interpreted as evidence that helpers/punishers who do not check the personal cost before deciding to help/punish return more of their endowment than helpers/punishers who do check the cost (i.e., that uncalculating helpers/punishers are more trustworthy than calculating helpers/punishers). Otherwise, there is no evidence for H14.1/H14.2a. H14.2b) A significant negative coefficient of decision time will be interpreted as evidence that punishers who take a short time to make their decision to punish return more of their endowment than punishers who are slower in making their decision (i.e., that uncalculating punishers are more trustworthy than calculating punishers). Otherwise, there is no evidence for H14.2b. If power requirements are not achieved, the results will be reported as suggestive, pending confirmation in future research. |
| Q15. Do uncalculated decisions around personal cost predict the actual trustworthiness of non-helpers/non-punishers? | Helping: H15.1) Non-helpers who checked the cost of helping will return significantly more of their endowment than non-helpers who did not check the cost. Punishing: H15.2a) Non-punishers who did not check the cost of punishing will return significantly more of their endowment than non-punishers who checked the cost. H15.2b) Fast deciding non-punishers will return significantly more of their endowment than slow deciding non-punishers. | Please refer to the Sampling plan in Methods for detail. H15.1 & H15.2a) As we do not know how many Player As will decide not to help/punish, the sample size for this model will be N = 1306 (653 Players A per condition in Experiment 1/2). H15.2b) As we do not know how many Player As will decide not to punish, the sample size for this model will be N = 1306 (653 Players A per condition in Experiment 3). | H15.1 & H15.2a) We will run a linear regression predicting the percentage of endowment returned by non-helpers/ non-punishers as a function of cost checking behaviour. H15.2b) We will run a linear regression predicting the percentage of endowment returned by non-punishers as a function of log-transformed punishing decision time, controlling for log-transformed general comprehension speed. | H15.1) A significant positive coefficient for cost checking behaviour (0 = did not check the cost, 1 = checked the cost) will be interpreted as evidence that non-helpers who checked the cost of helping return more of their endowment compared to non-helpers who did not check the cost of helping (i.e., calculating non-helpers are more trustworthy than uncalculating non-helpers). Otherwise, there is no evidence for H15.1. H15.2a) A significant negative coefficient for cost checking behaviour (0 = did not check the cost, 1 = checked the cost) will be interpreted as evidence that non-punishers who did not check the cost of punishing return more of their endowment than non-punishers who checked the cost (i.e., uncalculating non-punishers are more trustworthy than calculating non-punishers). Otherwise, there is no evidence for H15.2a. H15.2b) A significant negative coefficient for log-transformed decision time will be interpreted as evidence that fast deciding non-punishers return more of their endowment than slow deciding non-punishers (i.e., uncalculating non-punishers are more trustworthy than calculating non-punishers). Otherwise, there is no evidence for H15.2b. If power requirements are not achieved, the results will be reported as suggestive, pending confirmation in future research. |
| Q16. Does the operationalisation of uncalculating behaviour differentially influence the actual trustworthiness of punishers in the context of personal cost? | H16) Punishers who check the personal cost of punishing will return significantly less of their endowment than punishers who take a long time to decide. | Please refer to the Sampling plan in Methods for detail. H16) As we do not know how many Players A will decide to punish, the sample size for this model will be up to N = 2612 (653 Players A per condition in Experiments 2 and 3). | H16) We will run a linear regression predicting the percentage of endowment returned by Players A as a function of experiment (decision time vs cost checking) and deliberation (uncalculated vs calculated), as well as the interaction between experiment and deliberation. | H16) A significant negative coefficient for the interaction between experiment (0 = decision time, 1 = cost checking) and deliberation (0 = uncalculated, 1 = calculated) will be interpreted as evidence that the effect of calculated vs uncalculated punishment on trustworthiness is stronger for cost checking than decision time. Otherwise, there is no evidence for H16. If power requirements are not achieved, the results will be reported as suggestive, pending confirmation in future research. |
| Q17. Does the operationalisation of uncalculating behaviour differentially influence the actual trustworthiness of non-punishers in the context of personal cost? | H17) Non-punishers who check the personal cost of punishing will return significantly less of their endowment than non-punishers who take a long time to decide. | Please refer to the Sampling plan in Methods for detail. H17) As we do not know how many Player As will decide not to punish, the sample size for this model will be up to N = 2612 (653 Players A per condition in Experiments 2 and 3). | H17) We will run a linear regression predicting the percentage of endowment returned by Players A as a function of experiment (decision time vs cost checking) and deliberation (uncalculated vs calculated), as well as the interaction between experiment and deliberation. | H17) A significant negative coefficient for the interaction between experiment (0 = decision time, 1 = cost checking) and deliberation (0 = uncalculated non-punishment, 1 = calculated non-punishment) will be interpreted as evidence that the effect of calculated vs uncalculated non-punishment on trustworthiness is stronger for cost checking than decision time. Otherwise, there is no evidence for H17. If power requirements are not achieved, the results will be reported as suggestive, pending confirmation in future research. |

**Table 1 (continued) | Design Table**

| Question / Primary Hypotheses | Hypothesis | Sampling plan (e.g. power analysis) | Analysis Plan | Interpretation given to different outcomes |
|---|---|---|---|---|
| Q18. Do uncalculated helping and punishing decisions differentially influence actual trustworthiness in the context of personal cost? | H18) Helpers who check the personal cost of helping will return significantly less of their endowment than punishers who check the personal cost of punishing. | Please refer to the Sampling plan in Methods for detail. H18) As we do not know how many Players A will decide to help/punish, the sample size for this model will be up to $N = 2612$ (653 Players A per condition in Experiments 1 and 2). | H18) We will run a linear regression predicting the percentage of endowment returned by Players A as a function of behaviour and deliberation, as well as the interaction between behaviour and deliberation. | H18) A significant negative coefficient for the interaction between behaviour (0 = punishing, 1 = helping) and deliberation (0 = uncalculated, 1 = calculated) will be interpreted as evidence that the effect of calculated vs uncalculated decisions on trustworthiness is stronger for helping than punishing. Otherwise, there is no evidence for H18. If power requirements are not achieved, the results will be reported as suggestive, pending confirmation in future research. |
| Q19. Do uncalculated decisions around target impact predict the actual trustworthiness of helpers/punishers? | Helping: H19.1) Helpers who did not check the impact of helping will return significantly more of their endowment than helpers who checked the impact of helping. Punishing: H19.2) Punishers who checked the impact of punishing will return significantly more of their endowment than punishers who did not check the impact of punishing. | Please refer to the Sampling plan in Methods for detail. H19.1 & H19.2) As we do not know how many Players A will decide to help/punish, the sample size for this model will be up to $N = 1306$ (653 Players A per condition in Experiment 4/5). | Here, both the observable and the hidden condition will be used, as we collect the data on Player A's decision process, even when Player B cannot observe it. H19.1 & H19.2) We will run a linear regression predicting the percentage returned as a function of the helpers/punishers impact checking decision. | H19.1) A significant negative coefficient of impact checking (0 = did not check the impact, 1 = checked the impact) will be interpreted as evidence that helpers who do not check the impact on a target before deciding to help return more of their endowment than helpers who do check the impact (i.e., that uncalculating helpers are more trustworthy than calculating helpers). Otherwise, there is no evidence for H19.1. H19.2) A significant positive coefficient of impact checking (0 = did not check the impact, 1 = checked the impact) will be interpreted as evidence that punishers who do check the impact of punishing on targets before deciding to punish return more of their endowment than punishers who do not check the impact (i.e., that calculating punishers are more trustworthy than uncalculating punishers). Otherwise, there is no evidence for H19.2. If power requirements are not achieved, the results will be reported as suggestive, pending confirmation in future research. |
| Q20. Do uncalculated decisions around target impact predict the actual trustworthiness of non-helpers/non-punishers? | Helping: H20.1) Non-helpers who checked the impact of helping will return significantly more of their endowment than non-helpers who did not check the impact. Punishing: H20.2) Non-punishers who checked the impact of punishing will return significantly more of their endowment than non-punishers who did not check the impact. | Please refer to the Sampling plan in Methods for details. H20.1 & H20.2) As we do not know how many Player As will decide not to punish/help, the sample size for this model will be $N = 1306$ (653 Players A per condition in Experiment 4/5). | H20.1 & H20.2) We will run a linear regression predicting the percentage of endowment returned by non-helpers/non-punishers as a function of target impact checking behaviour. | H20.1 & H20.2) A significant positive coefficient for impact checking behaviour (0 = did not check the impact, 1 = checked the impact) will be interpreted as evidence that non-helpers/non-punishers who checked the impact of helping/punishing return more of their endowment compared to non-helpers/non-punishers who did not check the impact (i.e., calculating non-helpers/non-punishers are more trustworthy than uncalculating non-helpers/non-punishers). Otherwise, there is no evidence for H20.1/H20.2. If power requirements are not achieved, the results will be reported as suggestive, pending confirmation in future research. |

**Table 2 | Nomenclature for studies investigating whether helping and punishment decisions signal trustworthiness**

| Study Identification | | | | |
|---|---|---|---|---|
| **Study 1**<br>**Personal Cost Deliberation** | | | **Study 2**<br>**Target Impact Deliberation** | |
| **1.1**<br>**Help** | **1.2**<br>**Punish** | | **2.1**<br>**Help** | **2.2**<br>**Punish** |
| cost checking (E1) | *1.2a*<br>cost checking (E2) | *1.2b*<br>decision time (E3) | impact checking (E4) | impact checking (E5) |

We recruited 1,306 Player A - Player B pairs for each of the five experiments above (i.e., each experiment contains 1,306 Player As and 1,306 Player Bs). In each experiment, half of the players were assigned to the observable decision process condition, while the other half was assigned to the hidden decision process condition.

Despite this ambiguity, we still expected fast punishers to be trusted more than slow punishers in the context of personal cost deliberation. This is because observers were informed that the only new information Player A received right before making their punishing decision was the cost of punishment to themselves. Observers should infer, therefore, that deliberation stems only from the consideration of this personal cost and not the impact to the target. In addition, to disambiguate personal costs from harm aversion, we set the minimum potential cost of helping or punishing to be £0.00 in Study 1. Therefore, we anticipated that observers would send more of their endowment to third-party punishers who made their decision quickly, using uncalculated third-party punishment as a signal of trustworthiness. Similarly, we believed that being slower in the decision to punish would reflect the punisher's conflict about whether paying the cost would be beneficial to themselves, rather than an additional consideration of whether harming the violator is the "right thing to do". Thus, we expected uncalculated (fast) punishers to return more money than calculated (slow) punishers.

Study 2 was designed to address some of the open questions raised by Study 1 – specifically, whether deliberating over the impact on targets is perceived more positively for punishment than for helping. For both helping and punishment, we expected participants to be sensitive to the reputational consequences of their behaviour, albeit in different ways. If they want to be evaluated positively by an observer, helpers should be less likely to check how much helping will impact targets, whereas punishers should be more likely to check how much their actions will harm another. In other words, unlike Study 1, calculated punishment now served as a signal of trustworthiness: punishers who deliberate about the impact to the target should be entrusted with more money by observers and should be more trustworthy, compared to punishers who do not deliberate in this way. For helpers, as in Study 1, we expected uncalculated decisions to signal trustworthiness.

We also had several secondary predictions pertaining to decisions *not* to help or punish that can further clarify when and why deliberating over social actions carries reputation consequences. Specifically, we were interested in whether non-punishers are evaluated differently to non-helpers – and to what extent deliberation moderates these perceptions.

To understand how non-helpers and non-punishers are perceived, we must consider the potential motives driving decisions not to help or punish others. One primary reason individuals may refrain from helping or punishing in this task is due to personal costs. Additionally, non-helpers may not be especially motivated to help others because they are antisocial or inequity averse (i.e., they do not want someone else to receive more than them). For punishment, individuals may also refrain because they are averse to harming others. To disambiguate personal costs from harm aversion, we set the minimum cost of helping or punishing to be £0.00 in Study 1 (the minimum impact of helping or punishing was set to £0.01, so that some impact of investing in help or punishment was guaranteed). This feature allowed us to make nuanced predictions about how non-helpers and non-punishers would be evaluated.

A decision not to help is likely to stem primarily from self-interest and, consequently, unhelpful individuals are generally not trusted by others[24,26]. Those who decide not to help without checking the cost to themselves (Experiment 1.1) or the impact on the target (Experiment 2.1) might be evaluated especially negatively as it indicates an unwillingness to help, even if helping might impose no personal cost (Study 1) and regardless of the potential benefit to the target (Study 2). Conversely, deliberation indicates that the individual at least considered helping before deciding not to. Thus, in general, uncalculated decisions not to help should be evaluated negatively by observers in both studies, and uncalculating non-helpers should be less trustworthy than calculating non-helpers.

The same is not true for punishment: refraining from punishing others could stem from self-interest or from harm aversion. The possibility for non-punishment to stem from harm-aversion may help to explain why non-punishers can sometimes be trusted as much as punishers[24]. In Study 1, we expected that non-punishers who do not consider the personal cost might be perceived as harm averse – individuals who would not punish even if it were free to do so. Here, non-punishers who do not check the cost of punishing might be perceived as (and should actually be) relatively trustworthy. Conversely, non-punishers that check the personal cost before refraining from punishment should be seen as less trustworthy, because the inference is that these decisions were driven by self-interest (i.e. the personal cost being too large) rather than harm aversion.

In Study 2, uncalculated non-punishers (those who did not check the impact of punishment on targets) might either be completely harm averse or might be unwilling to pay the personal cost associated with punishing. Calculated non-punishers (those who *did* check the impact of punishment) by contrast, are those who might be willing to incur the personal cost of punishing but who wanted to know what impact this would have on the target before doing so. For uncalculated non-punishment to be perceived as more trustworthy than calculated non-punishment, participants would need to believe the target deserved no punishment. However, this would also imply tacit acceptance of the behaviour exhibited by Player 2 (returning nothing after their partner entrusted them with their entire endowment). Since both Players A and Players B knew they would subsequently be playing a Trust Game together, attitudes towards the target and what is considered acceptable behaviour in a Trust Game are relevant. As such, we expected calculated non-punishers to be perceived as and actually be more trustworthy than uncalculated non-punishers.

Given that the motives for actions are somewhat more transparent than those for non-actions, and that we expected incurring a cost to punish the cheat or help the cheated to be seen as more prosocial than doing nothing, we anticipated that deliberation (both of personal cost and target impact) would have a more substantial impact on trust and trustworthiness when individuals chose to punish or help than when they did not. Finally, we again expected non-punishers' decision speed to have a weaker effect on trust and trustworthiness than cost-checking decisions.

We must note, however, that predictions regarding the trustworthiness of punishers/helpers and non-punishers/non-helpers are considered exploratory, as we did not know whether they would achieve 95% power.

See Table 1 for more detailed descriptions of all hypotheses, and Fig. 1 for a visualisation thereof.

## Methods
### Design
We conducted experiments of highly similar designs to investigate different operationalisations of each calculating behaviour (i.e., (i) checking the cost or impact of punishment or helping, and (ii) punishing cost decision time). Each experiment recruited separate sets of both Players A and Players B, and had two conditions: decision process hidden or decision process observable, with a between-subjects design. The studies were built in Qualtrics (www.qualtrics.com) and consisted of two-stage, incentivized, anonymous economic games (see Fig. 2 for a visualisation of the study design). Player A made decisions during both games, whilst Players B only made decisions during the second game. Prior to making any decisions, Players A and

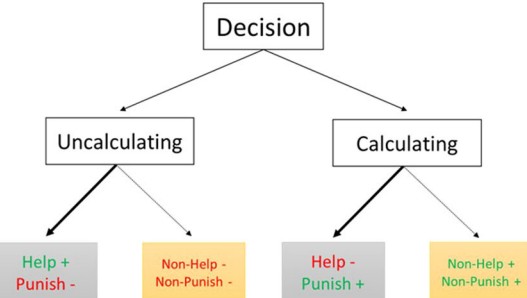

**Fig. 1 | Hypotheses for Study 1 (personal cost deliberation) and Study 2 (target impact deliberation).** In both studies, participants made an uncalculated or calculated helping/punishing decision. Lower boxes indicate our expectations regarding whether a calculated or uncalculated decision is associated with a comparatively higher level of trust and trustworthiness. Green text and plus signs indicate our expectations of increased trust and trustworthiness, while red text and minus signs indicate our expectations of decreased trust and trustworthiness. This figure demonstrates our expectation that uncalculated helping signals trustworthiness in the same way across studies, while we expected uncalculated punishment to be similar to uncalculated helping when personal cost is deliberated, but to differ when considering the impact on targets.

Players B read the instructions for both games to ensure that they could make informed decisions and comprehension could be assessed.

The following describes the procedure for Study 1 in a punishing context. We outline any differences for the helping context below.

**Stage 1**. Stage 1 is a standard Trust Game that was observed by both Player A and Player B. The outcome of a trust game played between 'Player 1' and 'Player 2' (who do not actually exist) was presented to Player A and B. Participants were told that Player 1 started with a £0.10 endowment and could choose how much of this endowment to send to Player 2, who started the trust game with no money. Participants were told that the amount entrusted was tripled by the experimenter and that Player 2 could then choose how much to return to Player 1. Participants were told that Player 1 and Player 2 already made their decisions: Player 1 sent their entire endowment (£0.10) to Player 2, who returned nothing, now leaving Player 1 with no money and Player 2 with £0.30.

Player A started with a £0.10 endowment and could then choose whether to use some of their endowment to 'punish' Player 2. Participants did not know exactly how much it would cost to punish Player 2, except that it would be somewhere between £0.00 and £0.10. Punishing always removed £0.15 from Player 2's bonus. We chose this amount as it leaves Player 2 with a bonus of £0.15, which would have been the 'fair' amount for Player 2 to receive in the trust game. Moreover, Player 2 still finishes Stage 1 with more than Player A, thereby reducing the potential that punishment is perceived as stemming from Player A wanting to have more than Player 2.

Players A additionally decided whether to make their punishing decision in a calculating or an uncalculating manner. The way this was operationalised varied between participants across experiments (Table 2).

**Cost-checking:** Player A chose whether to check the cost of punishing Player 2 before making their punishing decision. The cost of punishment was always £0.05.

**Decision time:** Player A was told that the cost of punishing is £0.05, after which they immediately made their punishing decision on the same page. The time spent on this page was recorded to assess their decision time.

**Stage 2**. In Stage 2, Players A and B participated in a trust game with each other. The trust games 'rules' were the same as in Stage 1. Player B started with a bonus of £0.10 and had to choose how much of their endowment (£0.00 - £0.10) to send to Player A. Any amount sent was tripled. Player A then chose what percentage of the endowment to return to Player B.

In the 'process observable' condition, Player B could condition their sending decision both on (i) Player A's Game 1 decision process (whether Player A chose to reveal the cost of punishing in Experiment 1.2a, or Player A's fast/slow decision time in Experiment 1.2b) and (ii) Player A's Game 1

decision (whether Player A punished Player 2). In the 'process hidden' condition, Player B could only condition their sending decision on Player A's Game 1 decision (whether they chose to punish Player 2). We employed the strategy method for both players: Player B decided how much to send to Player A who engaged in all possible combinations of punishing decisions and (depending on the condition) processes, without knowing what Player A did. Similarly, Player A decided what percentage of the amount they received from Player B to return, without knowing how much Player B sent. Participants were told that the choice that matched the decision the other player made would determine their payoff.

All participants were asked several comprehension questions, primarily to assess their understanding of the incentive structure of both games. In addition to Player A decision time, we recorded the time spent on the three pages with comprehension questions in Experiment 1.2b. These recordings serve as a control for general comprehension and reading speed.

After completing the data collection, we randomly matched pairs of Players A and Players B who participated in the same experiment and condition. The decisions participants made during the study then determined their bonus payments.

**Differences across studies and contexts**. The procedure across punishing and helping contexts was identical, except that instead of choosing whether to punish Player 2, Player A decided whether to use some of their endowment to help Player 1.

Whereas Study 1 explored how participants respond to an unknown personal cost of helping or punishing, Study 2 explored how participants respond to the unknown impact of helping or punishing another. In Study 2 participants were told that the personal cost of punishment (or helping) is £0.05 but they did not know exactly how much punishing would remove from Player 2 (or helping would benefit Player 1), except that it would be somewhere between £0.01 and £0.30. The maximum impact of £0.30 in Study 2 is equivalent to the maximum personal cost of £0.10 in Study 1: £0.10 was the entire endowment of the helper/punisher in Study 1, whereas £0.30 was the entire endowment of the target (in the punishment condition) in Study 2. Participants were informed that the minimum potential impact of helping/punishing a target is £0.01 because £0.00 would indicate no punishment or no help. When the impact was revealed, punishing still removed £0.15 from the target, and helping still delivered £0.15 to the target, just as in Study 1.

The procedure for Stage 2 only differed in that the Stage 1 procedure influenced what decisions and decision processes Players B could condition their sending decision on (i.e., whether it was a punishing or helping decision and whether it centered around personal cost checking or impact checking).

**Fig. 2 | Illustration of our two-stage experimental design investigating uncalculated punishment and helping in Studies 1 and 2, for both checking behaviour and decision speed.** In Game 1 Player A could pay a cost to punish/help another player. Player A decided (i) whether to make their decision in a calculated or uncalculated way (operationalised via their cost-checking (Experiments 1.1 & 1.2a) or impact-checking (Experiments 2.1 & 2.2) decisions, or their decision time (Experiment 1.2b), and (ii) whether to punish/help. In Game 2, Player B decided how much to send Player A (indicating how much they trust Player A), and Player A decided how much to return to Player B (indicating how trustworthy Player A is). There were two conditions in all experiments: in the process observable condition, Player B could base their sending decisions both on Player A's decision process (i.e., their checking decision or decision time) as well as Player A's punishing/helping decision, whilst in the process hidden condition Player B could only make their decisions based on Player A's punishing/helping decision.

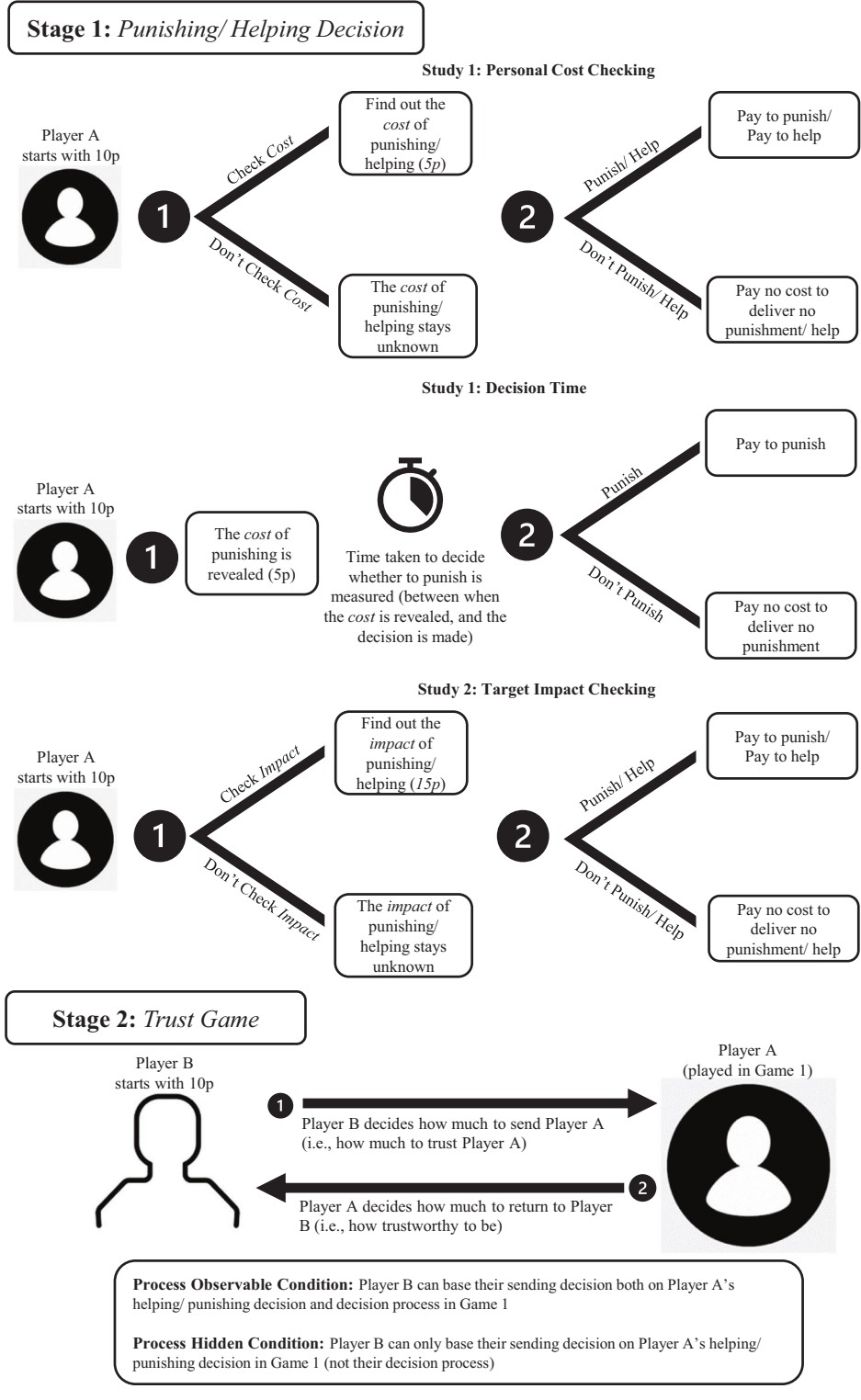

The procedure and instructions as seen by participants can be viewed in the supplementary information under "Supplementary Methods".

## Sampling plan

**Power Analysis.** Our power calculation was conducted in R[38] using the package 'pwr'[39] with the 'pwr.f2.test' function. We used a power of 0.95 with a 0.05 significance level and a one numerator degree of freedom ($u$, the number of coefficients in the model without the intercept). While estimating the required sample size, we referred to Jordan et al.'s[26]

supplementary materials for effect sizes, but specific effect sizes were not explicitly mentioned. We acknowledge that the available coefficients in their supplementary materials vary considerably, but generally produced small to medium effect sizes. As their study closely matches our experimental design, procedure, and research questions, we used an effect size of $f^2 = 0.02$ in our power analysis. According to Cohen's guidelines[40], $f^2 \geq 0.02$ represents a small effect size[41]. Because our main interests focussed on third-party punishment rather than helping, we expected to find similar or smaller effect sizes. Nevertheless, we must acknowledge

that our choice of $f^2 = 0.02$ might be considered a heuristic approximation rather than a precise estimation based on a formal inspection of Jordan et al.'s results. Based on the model with the highest number of predictors (as $n = v + p$, with p being the number of predictors including the intercept and v the degrees of freedom for the denominator), a sample size of 653 would be needed. However, as each of the models upon which this calculation was based involves either a Player A or a Player B participant, taking part in one of five experiments, in either the process hidden or in the process observable condition, a sample size of 13,060 (i.e., 1,306 Player A - Player B pairs per experiment) was needed. As we could not predict how many participants would decide to punish/ help or not punish/ help, it was not possible to ascertain before data collection whether 95% power would be achieved for all analyses. Results that did not meet the power requirements are therefore interpreted as suggestive, pending confirmation in future research.

**Participants**. We ran our experiments on Prolific (https://prolific.co/). Participants were invited to take part if they previously indicated on Prolific that they (i) are aged 18 or above, (ii) are from the UK, so that the currency specifications are familiar, (iii) are fluent in English, (iv) have the maximum approval rate of 100, and (v) selected "Yes, I would be comfortable to take part in such a study" to the question "Would you be happy to take part in a study where you are intentionally given inaccurate information about other participants and the study? You would be debriefed after the study". To avoid participants taking part in more than one experiment, we launched the experiments in sequence and allowed only new participants to take part. As preregistered, we lowered the approval rate to 97, in one-unit increments, as we did not reach enough participants with the maximum approval rate of 100.

**Data exclusion**. We used the "force response" feature in Qualtrics to ensure that we did not receive incomplete responses. As in Jordan et al.[26], responses by participants who failed more than one attention check were still included in the analyses. However, we re-ran the same analyses excluding those who failed more than one attention check, and reported this version when it led to significant differences in results. Any duplicate responses were removed.

## Reporting summary
Further information on research design is available in the Nature Portfolio Reporting Summary linked to this article.

## Analysis plan
Our analyses were conducted in R[38], and all analytical decisions for our hypotheses were independent of each other. For hypotheses in which we predicted cost-checking or impact-checking decisions (binary variables), we ran a logistic regression. For all other hypotheses, we used linear regressions, as they predict decision speed, sending decisions, or returning decisions (continuous variables). Decision speed was a continuous variable when returning decisions were predicted, but a dummy variable (median split of relatively fast or slow) was used when predicting sending decisions. For ease of interpretation, the measure for endowment sent was transformed from an absolute value (pence) sent to the percentage of endowment sent. The return measure was not transformed, as Players A already indicated what percentage, rather than what absolute value, they wished to return. In instances where Player B made sending decisions based on Player A's decision process, analyses were restricted to the process observable condition, as Player B did not know Player A's decision processes in the process hidden condition. Due to Player B making multiple sending decisions in each of these analyses (based on the possible decisions made by Player A during the first stage), each sending decision was treated as an observation, and robust standard errors were clustered on participant ID to account for the non-independence of repeated observations from the same participant (i.e., two observations per participant for decision process hidden, and four observations for decision process observable conditions). To accomplish

this, we utilized the lmtest package[42], employing the functions coeftest() with the argument vcov = vcovCL to specify the use of the sandwich estimator, and coefci(). As data was collected on Player A's decision process in both conditions (even when Player B could not observe it), data from both the observable and hidden conditions were used for analyses of returning decisions. As variance in decision time could also be caused by general comprehension ability or reading speed, rather than solely the time taken to reach a punishing decision, we included a control for general comprehension and reading speed when Player A's decision time is an independent variable. General comprehension and reading speed was operationalised as the natural log-transformed sum of time spent on the three comprehension question pages. All reported coefficients are unstandardised. For more detail on individual analysis methods for our hypotheses, see Table 1.

## Preregistered exploratory analyses
As we could not know how many participants would decide to punish/help or not punish/ help, it was impossible to ascertain before data collection whether 95% power would be achieved for all analyses in which Player A's returning decisions are predicted. Hypotheses H14.1 to H20.2 are therefore considered exploratory analyses. Results that do not meet the power requirements are interpreted as suggestive, pending confirmation in future research. This applied to hypotheses H14.2a, H14.2b, H15.1, H19.2, and H20.1, as they did not meet power requirements.

## Bayesian analyses
In addition to our frequentist analyses, we conducted equivalent Bayesian analyses to assess the evidence for each hypothesis compared to the null hypothesis. We used the BayesFactor package[43] with the lmBF function for linear regressions with return decisions as the response variable. For the logistic regressions and linear regressions with sending decisions as the response variable, we used the brm(), bridge_sampler(), and bayes_factor() functions from the brms package[44], which is better suited to handle the repeated observations from Player Bs. For analyses with repeat observations, we fit mixed models in brm() with ID as a random effect.

We constructed effect priors that are zero-centred t-distribution priors with 4 degrees of freedom. The prior width was designed such that only one-third of the prior mass on each side of zero is larger than the desired effect (i.e., the relevant coefficient observed in Jordan et al.[26]). Specifically, for any desired effect, one-third of the prior mass on each side of zero was more extreme than the absolute value of the desired effect. The total prior mass smaller than the desired effect was calculated as $0.5 + 0.5 * 2/3 = 0.83333$ (e.g. assuming an effect of 5.6 would lead to an effect prior with scale of 5.09). To achieve this, we used the below R code to calculate the scale of the prior width for a given desired effect (e.g. 5.6):

```
desiredEffect <- 5.6
myt <- function(x) { abs(extraDistr::qlst(0.5 + 0.5 * 2/3, df = 4, mu = 0, sigma = x) - desiredEffect) }
calc_scale <- optimize(myt, interval = c(0, 20)) prior_width_scale <- calc_scale$minimum.
```

The defined function "myt" calculates the absolute difference between the desired effect size (in this example 5.6) and the quantile of the t-distribution with 4 degrees of freedom and zero mean, corresponding to the prior mass of $0.5 + 0.5*2/3$. The "optimize" function in R then finds the value of the scale parameter that minimizes the absolute difference between the observed effect size and the quantile of the t-distribution. The resulting prior_width_scale value is what we used as the width of our prior distributions.

We chose this specification because of previous research from Jordan et al.[26] indicates effect sizes may generally be small, and as we investigated punishment as well as helping, effect sizes in punishing contexts may be smaller still. However, we still allowed for the possibility that we could sometimes find larger effects.

All other priors were weakly informative, using a zero-centred t-prior with 4 degrees of freedom and a scale of 10. We chose these weakly

informative priors to allow for some flexibility in the effect size estimates while still constraining them to reasonable values. The choice of 4 degrees of freedom and a scale of 10 reflects our prior belief that the effect size was unlikely to be very large, but may occasionally have been larger than expected.

To ensure that our prior effect size was appropriate, we set rscalefixed = 0.5 when using the BayesFactor package, as this is the smallest recommended prior to the effect size[45].

For hypotheses investigating the effect of deliberation on trust and trustworthiness, based on whether it is measured through cost checking or decision time, or takes place in the context of helping or punishing, we could not directly rely on equivalent coefficients from previous research to set priors as we did above. However, as we expected small effects, we set rscalefixed = 0.5 here as well, and to be conservative used the smallest interaction effect found in Jordan et al. to calculate the prior scale for analyses in which sending decisions are the response variable.

We also conducted sensitivity analyses for each Bayes factor test by conducting two additional analyses: one with a prior scale of 0.5 times the original value and one with a prior scale of 1.5 times the original value. We report the results of these sensitivity analyses in our supplementary materials, unless they changed the direction of the Bayes factor, in which case they are reported in the main text.

In evaluating the strength of evidence for or against the alternative hypothesis compared to the null hypothesis, we used common decision heuristics[46,47] and considered Bayes factors of 3 as weak evidence in favour of the alternative hypothesis, and Bayes factors of one-third as weak evidence in favour of the null hypothesis over the alternative hypothesis. Bayes factors of 10 or more were considered substantial evidence for the alternative hypothesis. Conversely, Bayes factors of one-tenth or less were considered substantial evidence for the null hypothesis. In cases where the Bayes factor fell between these thresholds, we concluded that the data provided no strong evidence for either the alternative or the null hypothesis and that more data were needed to draw a conclusive inference.

### Protocol Registration
The Stage 1 protocol for this Registered Report was accepted in principle on 13th November 2023. The protocol, as accepted by the journal, can be found at https://doi.org/10.6084/m9.figshare.24559462.v1.

### Deviations from Stage 1 protocol
Due to some participants starting but not finishing the experiment, some condition cells were unbalanced. We therefore recruited an additional 13 participants (five each in Experiments 1 and 4, and three in Experiment 2) to ensure that each condition reached the preregistered number of participants, bringing the total sample size to 13073 rather than 13060. Originally, we planned to run Bayesian analyses for H9.2b with lmBF(). However, as the function currently does not allow for models containing both continuous and categorical predictors, those models were fit with brm() as specified in the Bayesian Analysis section instead. Lastly, to maintain consistency with our registered analyses, we incorporated additional analyses centring around non-action in the comparison between help and punishment, ensuring comprehensive coverage and completeness across all hypotheses.

### Results
Median completion time for experiments ranged between six and seven minutes. Demographics were similar across experiments. Participants in Experiment 1 were aged between 18-80 years ($M = 39.45$, $SD = 12.51$) with 1381 women and 1207 men (17 identified as 'other' and 7 preferred not to say). In Experiment 2, participants were aged between 18-79 years ($M = 39.16$, $SD = 12.36$) with 1519 women and 1071 men (18 identified as 'other' and 7 preferred not to say). In Experiment 3, participants were predominantly women (1598 women, 990 men, 22 'other', and 2 preferred not to say) and were aged between 18 and 91 years ($M = 39.7$, $SD = 12.65$). 63% of participants in Experiment 4 were women (1645 women, 949 men,

21 'other', 2 preferred not to say) with a mean age of 37 years ($SD = 11.79$; range: 18-80 years). In Experiment 5, the average age was 38 years ($SD = 12.45$; range: 18-78 years), and 61% of participants were women (1581 women, 1002 men, 20 'other', and 9 preferred not to say). Data largely conformed to the assumptions of the tests used, but decision time was heavily skewed. As pre-registered, punishing decision time was therefore natural-log transformed.

### Preregistered Primary Hypotheses: Deliberation over help and punishment as a signal of trustworthiness
Our main prediction was that people would strategically adjust their decision-making process to gain reputational benefits. When considering the personal cost of helping (Exp. 1) and punishment (Exp. 2 & Exp. 3), as well as when considering the impact of helping (Exp. 4), we expected Players A to be more likely to make uncalculating decisions (by deciding quickly or without checking the personal cost or target impact) when their decision process was observed by others (and could therefore potentially confer reputation benefits), compared to when their decision process was hidden. We predicted the opposite when participants considered the impact of punishment on a target (Exp. 5): as punishment is a harmful act, we expected Players A to make calculating decisions (by checking the impact that punishment would have on the target) when their decision process was observed. See Fig. 3 for a visualisation of results and Table 2 for the nomenclature of experiments.

In line with predictions, participants were around half as likely (odds ratio (OR) = 0.64, 95% confidence interval (CI) = [0.50, 0.82]) to check the personal cost of helping when their decision process was observable than when it was hidden (Exp. 1, H1.1). Specifically, 78% of Players A checked the cost of helping in the decision process hidden condition, whereas only 69% did so in the decision process observable condition ($b = -0.44$, 95% CI = [−0.69, −0.19], $p < 0.001$, BF = 71.29).

Similarly, the odds of checking the personal cost of punishing decreased by around 41% (OR = 0.59, 95% CI = [0.47, 0.73]) when the participants' decision process was observable compared to hidden (Exp. 2, H1.2a). Specifically, 67% of Players A checked the cost of punishing in the decision process hidden condition, whereas only 54% did so in the decision process observable condition ($b = -0.53$, 95% CI = [−0.76, −0.31], $p < 0.001$, BF = 6972.38).

Players A also made significantly faster punishing decisions when their decision process was observable ($M = 2.11$ log-seconds, 95% CI = [2.06, 2.15]) compared to hidden ($M = 2.19$ log-seconds, 95% CI = [2.14, 2.23]), with a 7.86% decrease in decision time in the observable condition ($t(1304) = -2.68$, $p = 0.007$, $b = -0.08$, 95% CI = [−0.14, −0.02], BF = 2.16) (Exp. 3, H1.2b).

In Study 2 (checking the impact of helping or punishing a target), results were less clear-cut. As predicted, participants were significantly more likely to check the impact of helping when their decision process was hidden (83%) compared to observable (78%) ($b = -0.34$, 95% CI = [−0.62, −0.06], $p = 0.016$, BF = 3.86; OR = 0.71, 95% CI = [0.54, 0.94]). Nevertheless, when only those who correctly responded to at least 7 out of 8 comprehension questions were included (hereafter referred to as participants with excellent comprehension), results were in the same direction but no longer significant (81% checked the impact when their decision process was hidden compared to 77% when it was observable; OR = 0.78 [0.51, 1.19]; $b = -0.24$, 95% CI = [−0.66, 0.18], $p = 0.26$, BF = 0.67) (Exp. 4, H5.1).

We expected that participants would be more likely to check the impact of punishment when their decision process was observed (Exp. 5, H5.2), but this prediction was not supported by the data (OR = 0.84 [0.66, 1.07]; $b = -0.17$, 95% CI = [−0.41, 0.07], $p = .16$, BF = 0.60). Indeed, when only participants with excellent comprehension were included, we found the opposite: 75% of Players A checked the impact of punishing in the decision process hidden condition, whereas only 64% did so in the decision process observable condition (OR = 0.62, 95% CI = [0.42, 0.89]; $b = -0.49$, 95% CI = [−0.86, −0.11], $p = 0.011$, BF = 5.89).

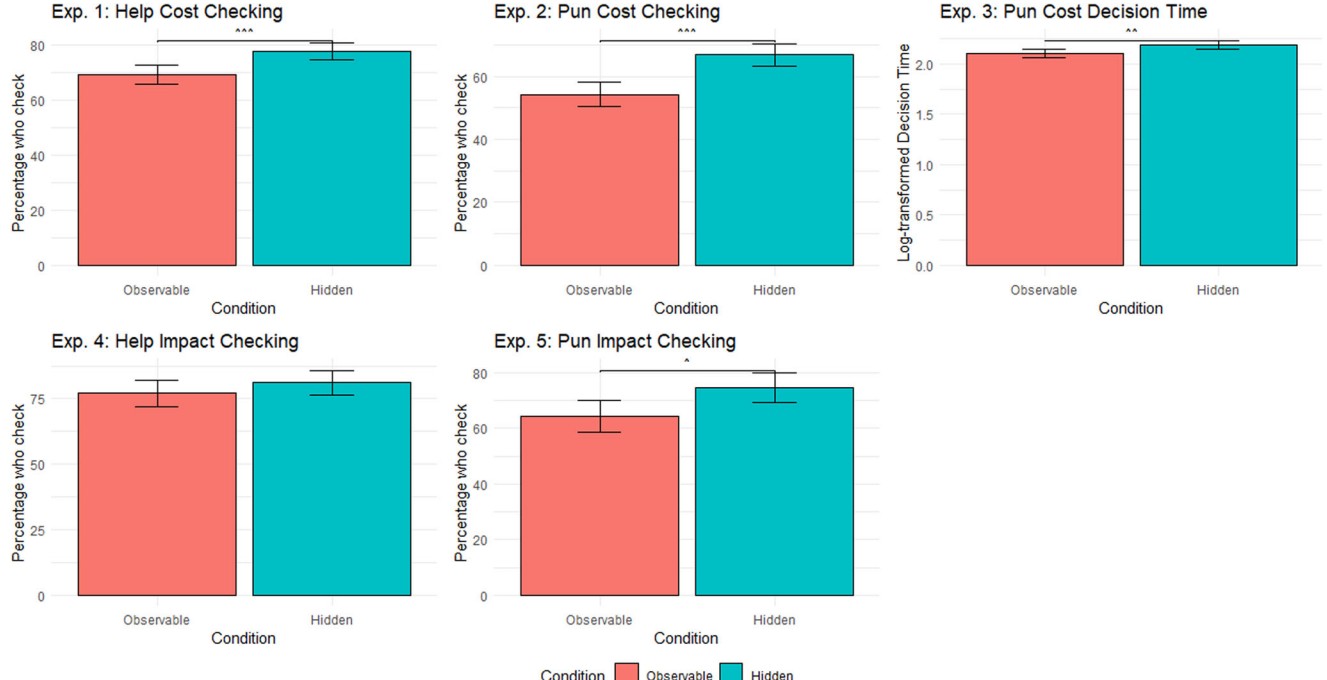

**Fig. 3 | Decision processes (uncalculating vs calculating) across all five experiments, in the decision process observable and hidden conditions.** Checking (versus not checking) the personal cost or target impact (Exp. 1, 2, 4 & 5), as well as taking a long (versus a short) time to decide (Exp. 3), reflect calculated decision making. Error Bars indicate 95% CI. Due to changes in significance levels, bar charts for Exp. 4 and Exp. 5 only include participants with excellent comprehension ($n = 1311$ for Exp. 1, $n = 1309$ for Exp. 2, n = 1306 for Exp. 3, $n = 534$ for Exp. 4, $n = 535$ for Exp. 5). Differences are significant for all but Exp. 4 (help impact checking).

### Preregistered Primary Hypotheses: The influence of deliberation over help and punishment on perceived trustworthiness

Next, we explored how helping and punishment decisions were interpreted by observers. We expected uncalculated help and punishment in the context of personal cost deliberation, as well as uncalculated help and calculated punishment in the context of target impact deliberation, to confer reputational benefits. Specifically, we expected observers to send helpers and punishers a higher percentage of their endowment in those situations, which we interpret as higher trust. See Fig. 4 for a visualisation of results.

Contrary to predictions, we found no statistically significant difference in the proportion of their endowment that observers sent to helpers who did not check the personal cost of helping ($M = 63.40\%$, $SD = 34.24$) than to helpers who checked the cost ($M = 60.57\%$, $SD = 33.33$) ($t(1304) = -1.52$, $p = 0.13$, $b = -2.83$, 95% CI = $[-6.50, 0.82]$, BF = 6.00) (Exp. 1, H2.1). Yet, while the preregistered frequentist statistics do not support H2.1 when all participants were included in the analysis, the preregistered Bayesian analysis, with a Bayes Factor > 3 indicates support for H2.1. Importantly, when only participants with excellent comprehension were included in the analysis, we found that observers sent a significantly higher proportion of their endowment to helpers who did not check the personal cost of helping ($M = 68.27\%$, $SD = 33.89$) than to helpers who checked the cost ($M = 62.65\%$, $SD = 33.73$) ($t(610) = -2.06$, $p = 0.04$, $b = -5.62$, 95% CI = $[-10.99, -0.26]$, BF = 8.97).

Our predictions that observers would send more money to punishers who made uncalculating decisions (when considering personal costs) were not supported. If anything, observers entrusted a higher proportion of their endowment to punishers who checked the personal cost of punishment ($M = 51.49\%$, $SD = 36.26$) than to those who did not ($M = 48.76\%$, $SD = 37.62$) (Exp. 2, H2.2a). However, this difference was statistically non-significant ($t(1304) = 1.33$, $p = 0.18$, $b = 2.73$, 95% CI = $[-1.28, 6.74]$, BF = 4.78). When calculating behaviour was operationalised in terms of decision time, observers sent more to relatively slow (more calculating) punishers ($M = 49.17\%$, $SD = 34.75$) than to relatively fast punishers ($M = 47.40\%$,

$SD = 36.80$) (H2.2b). Again, this result was not statistically significant ($t(1304) = -0.90$, $p = 0.37$, $b = -1.78$, 95% CI = $[-5.66, 2.11]$, BF = 0.39).

We expected that helpers who did not check the impact of helping behaviour would be trusted more by observers. Although observers sent a higher percentage of their endowment to helpers who did not check the impact ($M = 63.12\%$, $SD = 32.39$) than to those who did ($M = 61.53\%$, $SD = 31.68$) (Exp. 4, H6.1), this result was statistically non-significant ($t(1304) = -0.90$, $p = 0.37$, $b = -1.60$, 95% CI = $[-5.07, 1.88]$, BF = 1.53).

Another unsupported prediction was that impact-checking punishers would be trusted more by observers. Although observers did send more of their endowment to punishers who checked the impact of punishing ($M = 48.45\%$, $SD = 34.93$) than to punishers who did not ($M = 45.54\%$, $SD = 35.01$) (Exp. 5, H6.2), this difference was also statistically non-significant ($t(1304) = 1.50$, $p = 0.13$, $b = 2.91$, 95% CI = $[-0.89, 6.71]$, BF = 4.31). While the preregistered frequentist statistics do not support H6.2, the preregistered Bayesian analysis, with a Bayes Factor > 3 indicates support for H6.2.

### Exploratory Preregistered Hypotheses: The influence of deliberation over help and punishment on trustworthiness

Next, we asked whether calculated/uncalculated help and punishment decisions reliably signalled trustworthiness (Fig. 5). We expected uncalculating helpers in both the personal cost (Exp. 1) and impact checking context (Exp. 4) to be more trustworthy than calculating helpers. Indeed, helpers who did not check the personal cost of helping returned significantly more of the endowment they were sent by observers ($M = 48.74\%$, $SD = 19.76$) than helpers who did check the personal cost ($M = 43.54\%$, $SD = 19.34$) ($t(1099) = -3.85$, $p < 0.001$, $b = -5.21$, 95% CI = $[-7.86, -2.55]$, BF = 97.33) (Exp. 1, H14.1). Similarly, helpers who did not check the impact of helping ($M = 48.23\%$, $SD = 19.04$) returned a higher percentage in the Trust Game than helpers who checked the impact of helping ($M = 45.40\%$, $SD = 18.57$), but this effect was statistically non-significant ($t(1138) = -1.96$, $p = 0.05$, $b = -2.83$, 95% CI = $[-5.67, 0.004]$, BF = 0.44) (Exp. 4, H19.1).

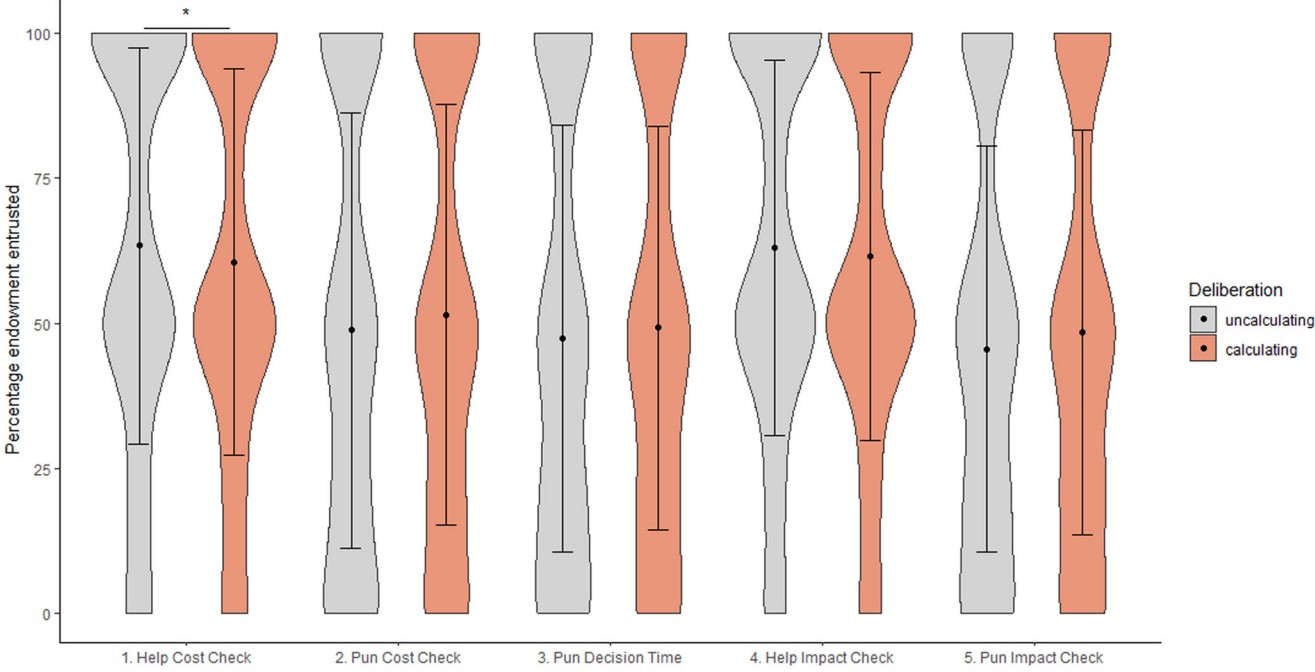

**Fig. 4 | Percentage of endowment sent to uncalculating and calculating helpers (Exp. 1 & Exp. 4) and punishers (Exp. 2, Exp. 3 & Exp. 5) by observers.** Checking (versus not checking) the personal cost or target impact, as well as taking a long (versus a short) time to decide reflects calculated decision-making. These data show percentage of endowment sent only to players who did help or punish others and not those who refrained from acting. The width of the violins indicates the distribution of observations, error bars indicate 95% CI, and dots represent the mean. Due to changes in significance levels, Exp. 1 (help cost checking) only includes participants with excellent comprehension (n = 612 for Exp. 1, n = 1306 for Exp. 2, n = 1306 for Exp. 3, n = 1306 for Exp. 4, n = 1306 for Exp. 5). Differences are significant for Exp. 1 (help cost checking).

We expected that punishers who made an uncalculating versus calculating decision in the context of personal cost (Exp. 2 and 3) would be more trustworthy. Conversely, for impact consideration, we predicted that punishers who made calculating decisions would be more trustworthy than punishers who made uncalculating decisions (Exp. 5). Our results did not support these predictions. Although punishers who did not check the personal cost of punishing returned more of the entrusted endowment ($M = 46.11\%$, $SD = 15.48$) than punishers who did check the personal cost ($M = 43.78\%$, $SD = 20.01$), this difference was not statistically significant ($t(506) = -1.04$, $p = 0.30$, $b = -2.33$, 95% CI = $[-6.76, 2.09]$, BF = 0.17) (Exp. 2, H14.2a). Conversely, when uncalculating decisions were operationalised as decision time, punishers who made slower (more calculating) decisions returned a slightly higher percentage than those who made faster (uncalculating) punishing decisions ($t(513) = 0.66$, $p = 0.51$, $b = 0.98$, 95% CI = $[-1.94, 3.90]$, BF = 0.14) (Exp. 3, H14.2b). This difference was not significant. Punishers who did not check the impact of punishing returned a lower percentage ($M = 38.48\%$, $SD = 22.53$) than punishers who did check the impact of punishing on the target ($M = 40.03\%$, $SD = 19.67$), $t(408) = 0.58$, $p = 0.56$, $b = 1.55$, 95% CI = $[-3.68, 6.78]$, BF = 0.12 (Exp. 5, H19.2). Although directionally in line with predictions, this difference too was non-significant.

It must be noted, that hypotheses H14.2a, H14.2b and H19.2 did not meet power requirements, therefore making their results suggestive, pending confirmation in future research. However, their Bayes Factor values indicate support for the null hypotheses (see Supplementary Table 1 under "Supplementary Notes 1" in the Supplementary Information for sensitivity analyses).

**Preregistered Primary Hypotheses: Trust and trustworthiness across experiments**

We expected uncalculated decision-making to differentially influence trust and trustworthiness across the experiments (Exp. 1-3) of Study 1 (personal cost). Firstly, for punishment, we predicted that deliberation would have a stronger influence on trust and trustworthiness when calculating behaviour was operationalised as cost-checking (Exp. 2) compared to slow decision time (Exp. 3). In addition, we expected deliberation to have a stronger effect on trust and trustworthiness in the context of helping compared to punishing (Exp. 1 vs Exp. 2).

However, the effect of calculated versus uncalculated punishment on trust was not stronger for cost checking than decision time ($t(2608) = 0.33$, $p = 0.74$, $b = 0.95$, 95% CI = $[-1.99, 3.88]$, BF = 0.10) (H3). The same was true for non-punishment ($t(2608) = -0.19$, $p = 0.85$, $b = -0.55$, 95% CI = $[-3.45, 2.35]$, BF = 0.75) (H10). Similarly, the effect of calculated versus uncalculated punishment on trustworthiness was not stronger for cost checking than decision time, $t(1020) = -1.27$, $p = 0.21$, $b = -3.44$, 95% CI = $[-8.78, 1.90]$, BF = 0.24 (H16). Again, the same was true for non-punishment: $t(1587) = -1.72$, $p = 0.09$, $b = -3.80$, 95% CI = $[-8.13, 0.54]$, BF = 0.33 (H17).

Observers trusted helpers significantly more than they trusted punishers ($t(2608) = 7.47$, $p < 0.001$, $b = 14.64$, 95% CI = $[10.74, 18.54]$). Moreover, trust was significantly influenced by the interaction between behaviour (helping versus punishing) and decision process (calculating versus uncalculating) ($t(2608) = -2.01$, $p = 0.04$, $b = -5.56$, 95% CI = $[-8.29, -2.83]$, BF = 140.10) (H4). Specifically, uncalculating punishers were trusted the least ($M = 48.76\%$, $SD = 37.62$), followed by calculating punishers ($M = 51.49\%$, $SD = 36.26$), calculating helpers ($M = 60.57\%$, $SD = 33.33$), and uncalculating helpers ($M = 63.40\%$, $SD = 34.24$). Uncalculating helpers were trusted significantly more than uncalculating punishers ($t(2608) = 7.47$, $p < 0.001$, $b = 14.64$) and calculating helpers were trusted significantly more than calculating punishers ($t(2608) = 4.64$, $p < 0.001$, $b = 9.08$).

This interaction was no longer significant when excluding those who failed more than one comprehension check ($t(1162) = -1.54$, $p = 0.12$, $b = -6.63$, 95% CI = $[-10.66, -2.60]$, BF = 20.22). While the preregistered frequentist statistics no longer support H4 when only participants with excellent comprehension were included, the preregistered Bayesian analysis,

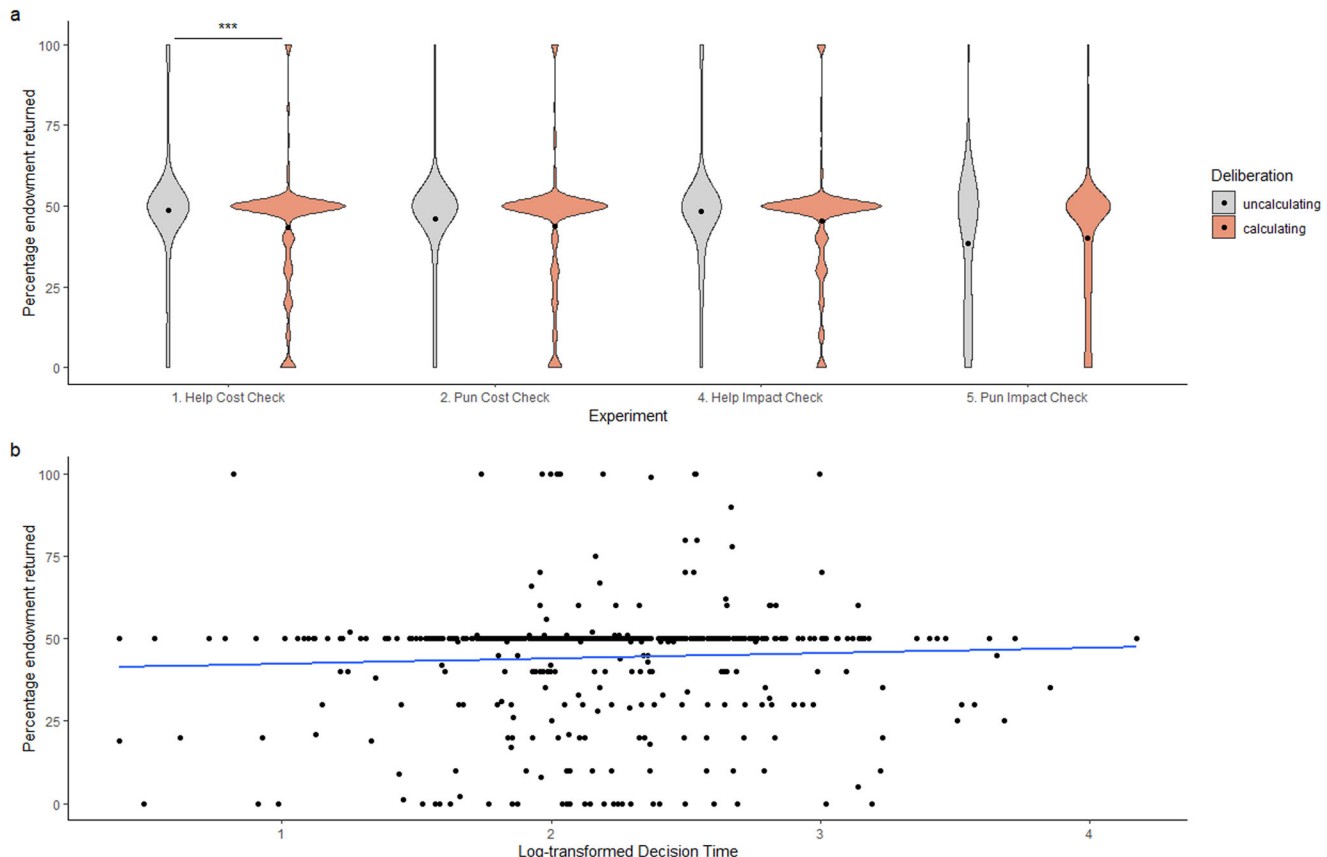

**Fig. 5 | Percentage of endowment returned to observers by uncalculating and calculating helpers (Exp. 1 & Exp. 4) and punishers (Exp. 2, Exp. 3 & Exp. 5) in the Trust Game.** Checking (versus not checking) the personal cost or target impact (Exp. 1, 2, 4 & 5; **a**), as well as taking a longer time to decide (Exp. 3; **b**) reflect calculated decision making. These data show the percentage of endowment returned only by players who did help or punish others and not those who refrained from acting. In **a** the width of the violins indicates the distribution of observations, error bars indicate 95% CI, and dots represent the mean. **b** shows a scatterplot with a regression line. Differences are significant for Exp. 1 (help cost checking). Participant numbers vary across experiments ($n = 1101$ for Exp. 1, $n = 508$ for Exp. 2, $n = 515$ for Exp. 3, $n = 1140$ for Exp. 4, $n = 410$ for Exp. 5).

with a Bayes Factor > 3 indicates support for H4. Observers still trusted helpers significantly more than punishers ($t(1605) = 4.28$, $p < 0.001$, $b = 13.00$, 95% CI = [6.87, 19.13]). Specifically, observers trusted uncalculating punishers the least ($M = 55.27\%$, $SD = 40.76$), increasing their levels of trust for calculating punishers ($M = 56.28\%$, $SD = 38.23$), calculating helpers ($M = 62.65\%$, $SD = 33.73$), and uncalculating helpers ($M = 68.27\%$, $SD = 33.89$).

There was also no evidence to suggest that the effect of calculated versus uncalculated decision-making on trustworthiness is stronger in helping compared to punishing contexts ($t(1605) = −1.09$, $p = 0.28$, $b = −2.88$, 95% CI = [−8.04, 2.29], BF = 0.17) (H18), and there was no significant difference in the trustworthiness of helpers and punishers ($t(1605) = 1.11$, $p = 0.27$, $b = 2.63$, 95% CI = [−2.01, 7.27]).

**Preregistered Secondary Hypotheses: The influence of deliberation over decisions not to help or punish on perceived trustworthiness**
Moreover, we had diverging expectations for how uncalculating decisions would be perceived when those decisions result in inaction rather than helping or punishing. We predicted that observers would send more to calculating than uncalculating non-helpers/non-punishers in Experiments 1, 4, and 5, but more to uncalculating than calculating non-punishers when personal cost is being considered (Exp. 2 and Exp. 3). However, none of these analyses were statistically significant.

Directionally in line with predictions, observers sent more of their endowment to non-helpers who checked the cost of helping ($M = 29.75\%$, $SD = 33.70$) than to non-helpers who did not check the cost of helping ($M = 28.81\%$, $SD = 34.38$) ($t(1304) = 0.50$, $p = 0.61$, $b = 0.95$, 95% CI = [−2.75, 4.64], BF = 0.68) (Exp.1, H7.1). Conversely, and again in line with predictions, in Experiment 2 (H7.2a) observers sent directionally less of their endowment to non-punishers who checked the personal cost of punishing ($M = 50.31\%$, $SD = 35.38$) than to non-punishers who did not check the cost of punishing ($M = 51.58\%$, $SD = 37.83$) ($t(1304) = −0.63$, $p = 0.53$, $b = −1.27$, 95% CI = [−5.25, 2.70], BF = 0.80). However, Experiment 3 (H7.2b) found that observers sent more of their endowment to relatively slow (calculating) non-punishers ($M = 49.36\%$, $SD = 37.18$) than to relatively fast (uncalculating) non-punishers ($M = 48.64\%$, $SD = 35.97$) ($t(1304) = 0.36$, $p = 0.72$, $b = 0.72$, 95% CI = [−3.25, 4.69], BF = 0.26). In Experiment 4 (H11.1) observers were again in line with predictions and sent more of their endowment to non-helpers who checked the impact of helping ($M = 32.43\%$, $SD = 34.22$) than to non-helpers who did not ($M = 29.71\%$, $SD = 33.17$) ($t(1304) = 1.46$, $p = 0.14$, $b = 2.73$, 95% CI = [−0.93, 6.38], BF = 5.05). In Experiment 5 (H11.2) observers sent similar amounts of their endowment to non-punishers who checked the impact of punishing ($M = 52.48\%$, $SD = 34.09$) and to non-punishers who did not ($M = 52.85\%$, $SD = 34.68$) ($t(1304) = −0.19$, $p = 0.85$, $b = −0.37$, 95% CI = [−4.10, 3.37], BF = 0.48).

**Exploratory Preregistered Hypotheses: The influence of deliberation over decisions not to help or punish on trustworthiness**
We also had diverging expectations for how uncalculating decisions would be associated with the actual trustworthiness of non-helpers and non-punishers. Specifically, we predicted that calculating non-punishers in the context of impact checking (Exp. 5) and calculating non-helpers in both the

context of impact (Exp. 4) and cost checking (Exp. 1) would return more than uncalculating non-helpers/non-punishers. In contrast, we expected uncalculating non-punishers to return more than calculating non-punishers in the context of personal cost deliberation (Exp. 2 & Exp. 3). All returning decisions for non-punishers and non-helpers were directionally in line with predictions.

In Experiment 1 (H15.1), non-helpers who checked the personal cost of helping returned more of their endowment ($M = 21.29\%$, $SD = 22.95$) than non-helpers who did not check the cost ($M = 15.69\%$, $SD = 22.88$) ($t(208) = 1.67$, $p = 0.10$, $b = 5.60$, 95% CI = $[-1.00, 12.19]$, BF = 0.56). However, this difference was non-significant. As predicted, in Experiment 2 (H15.2a), non-punishers who did not check the cost of punishing ($M = 42.51\%$, $SD = 21.65$) returned significantly more of their endowment than non-punishers who checked the cost ($M = 38.51\%$, $SD = 22.42$) ($t(799) = -2.57$, $p = 0.01$, $b = -4.0$, 95% CI = $[-7.06, -0.94]$, BF = 1.98). In Experiment 3 (H15.2b) uncalculating (faster) non-punishers again returned more of their endowment than calculating (slower) non-punishers, but this was not significant ($t(787) = -0.29$, $p = 0.77$, $b = -0.43$, 95% CI = $[-3.33, 2.47]$, BF = 0.01). In Experiment 4 (H20.1) non-helpers who checked the impact of helping ($M = 25.44\%$, $SD = 23.84$) returned significantly more of their endowment than non-helpers who did not check the impact ($M = 17.24\%$, $SD = 26.56$) ($t(169) = 1.99$, $p = 0.48$, $b = 8.21$, 95% CI = $[0.06, 16.35]$, BF = 1.02). However, the difference (calculating non-helper: 23.40% ($SD = 24.55$), uncalculating non-helper: 16.00% ($SD = 24.11$)) was no longer statistically significant when only those with excellent comprehension were included ($t(58) = 1.11$, $p = 0.27$, $b = 7.40$, 95% CI = $[-5.98, 20.78]$, BF = 0.44). Finally, in Experiment 5 (H20.2) both non-punishers who checked the impact of punishing ($M = 38.39\%$, $SD = 22.74$) and non-punishers who did not check the impact ($M = 38.26\%$, $SD = 24.68$) returned around 38% of their endowment ($t(822) = 0.08$, $p = 0.94$, $b = 0.13$, 95% CI = $[-3.17, 3.43]$, BF = 0.08).

It must be noted that power requirements were not met for hypotheses H15.1 (Exp. 1) and H20.1 (Exp. 4), making those results suggestive, pending confirmation in future research.

### Preregistered Secondary Hypotheses: The influence of deliberation on perceived and actual trustworthiness when decisions result in helping or punishing versus inaction

Lastly, for all experiments we predicted that the effect of uncalculating behaviour on trust and trustworthiness would be larger for action than inaction, meaning that deliberation would more strongly influence sending and returning decisions when Player A decided to help/punish compared to when Player A decided *not* to help/punish.

However, for sending decisions this was not the case in Experiment 1 (H8.1; $t(2608) = -1.43$, $p = 0.15$, $b = -3.78$, 95% CI = $[-8.99, 1.42]$, BF = 1.16), Experiment 2 (H8.2a; $t(2608) = 1.39$, $p = 0.17$, $b = 4.0$, 95% CI = $[-1.65, 9.64]$, BF = 1.17), Experiment 3 (H8.2b; $t(2608) = -0.88$, $p = 0.38$, $b = -2.50$, 95% CI = $[-5.40, 0.40]$, BF = 0.28), or Experiment 5 (H12.2; $t(2608) = 1.21$, $p = 0.23$, $b = 3.28$, 95% CI = $[-2.05, 8.60]$, BF = 0.79). Yet, when only participants with excellent comprehension were included, there was a significant interaction between deliberation and helping decision in Experiment 4 (H12.1; $t(1060) = -2.06$, $p = 0.04$, $b = -7.97$, 95% CI = $[-15.54, -0.40]$, BF = 5.89). Specifically, observers entrusted uncalculating non-helpers with only 21.73% ($SD = 29.24$) of their endowment, and calculating non-helpers with 26.77% ($SD = 31.86$) of their endowment. Helpers were sent more than twice as much: calculating helpers were entrusted with 61.20% ($SD = 32.02$) and uncalculating helpers received the most with 64.14% ($SD = 32.72$). Hereby, the differences between uncalculating helpers versus uncalculating non-helpers ($t(1060) = 15.53$, $p < 0.001$, $b = 42.41$) and calculating helpers versus calculating non-helpers ($t(1060) = 12.61$, $p < 0.001$, $b = 34.44$) were statistically significant. Moreover, in Experiment 1 there was a main effect for helping ($t(2608) = 18.43$, $p < 0.001$, $b = 34.59$, 95% CI = $[30.87, 38.32]$), as observers entrusted more than twice as much to helpers than to non-helpers, and in Experiment 5 observers sent significantly less to punishers than to non-punishers ($t(2608) = -3.81$, $p = 0.0001$, $b = -7.30$, 95% CI = $[-11.09, -3.52]$).

Furthermore, we found no evidence to suggest that deliberation had a larger effect on actual trustworthiness for punishers compared to non-punishers, as the interaction effects were non-significant in Experiment 2 (H9.2a; $t(1305) = 0.58$, $p = 0.56$, $b = 1.67$, 95% CI = $[-3.96, 7.29]$, BF = 0.14), Experiment 3 (H9.2b; $t(1301) = 0.88$, $p = 0.38$, $b = 1.85$, 95% CI = $[-2.29, 6.00]$, BF = 0.3) and Experiment 5 (H13.2; $t(1302) = 0.42$, $p = 0.67$, $b = 1.42$, 95% CI = $[-5.17, 8.01]$, BF = 0.13). For participants with excellent comprehension there were, however, main effects for both punishing ($t(615) = 2.13$, $p = 0.03$, $b = 6.33$) and checking ($t(615) = -2.30$, $p = 0.02$, $b = -4.49$) in Experiment 2, with punishers and uncalculating decision-makers returning significantly more than non-punishers and calculating decision makers.

In Experiment 1 (H9.1) the effect of uncalculating decision-making on trustworthiness was significantly larger when Players A decided to help compared to when Player A decided not to help ($t(1307) = -3.34$, p < 0.001, $b = -10.81$, 95% CI = $[-17.16, -4.45]$, BF = 18.29). In line with predictions, non-helpers who did not check the personal cost of helping were the least trustworthy, returning only an average of 15.69% ($SD = 22.88$), whilst cost-checking non-helpers returned 21.29% ($SD = 22.95$). Cost-checking helpers were substantially more trustworthy, returning an average of 43.54% ($SD = 19.34$), whilst helpers who did not check the personal cost returned the most, with an average of 48.74% ($SD = 19.34$). Post hoc tests on the estimated marginal means, accounting for multiple comparisons with the multivariate t-test (mvt) adjustment, revealed significant differences between uncalculating helpers and uncalculating non-helpers ($t(1307) = 12.40$, $p < 0.001$, $b = 33.05$), calculating helpers and calculating non-helpers ($t(1307) = 12.11$, $p < 0.001$, $b = 22.25$) as well as calculating helpers and uncalculating helpers ($t(1307) = -3.74$, $p < 0.001$, $b = -5.21$), but not for calculating non-helpers and uncalculating non-helpers ($t(1307) = 1.92$, $p = 0.18$, $b = 5.60$).

Conversely, and against predictions, in Experiment 4 (H13.1) the effect of uncalculating decision-making on trustworthiness was significantly *smaller* when Player A decided to help compared to when Player A decided not to help ($t(1307) = -3.07$, $p = 0.002$, $b = 11.04$, 95% CI = $[-18.10, -3.98]$, BF = 8.59). Nevertheless, in line with predictions, non-helpers who did not check the impact of helping were the least trustworthy, returning only an average of 17.24% ($SD = 26.56$) of the endowment observers entrusted them with, whilst calculating non-helpers returned an average of 25.44% ($SD = 23.84$), calculating helpers an average of 45.40% ($SD = 18.57$) and uncalculating helpers an average of 48.23% ($SD = 19.04$). Hereby, the differences between uncalculating helpers and uncalculating non-helpers ($t(1307) = 10.13$, $p < 0.001$, $b = 31.00$), calculating helpers and calculating non-helpers ($t(1307) = 10.54$, $p < 0.001$, $b = 19.96$), as well as uncalculating non-helpers and calculating non-helpers ($t(1307) = 2.51$, $p = 0.04$, $b = 8.21$) were statistically significant.

### Exploratory Unregistered Analyses: The influence of deliberation on trust and trustworthiness for non-helpers versus non-punishers

To provide a comprehensive perspective, a final unregistered analysis tested whether the effect of uncalculating behaviour on trust and trustworthiness differs for non-punishers compared to non-helpers. For trust there was no interaction between deliberation and behaviour ($t(2608) = 0.80$, $p = 0.42$, $b = 2.22$, 95% CI = $[-0.64, 5.08]$, BF = 0.98), nor a significant main effect for deliberation ($t(2608) = -0.65$, $p = 0.52$, $b = -1.27$, 95% CI = $[-3.39, 0.85]$,). However, observers sent significantly more of their endowment to non-punishers than to non-helpers ($t(2608) = -11.64$, $p < 0.001$, $b = -22.77$, 95% CI = $[-26.70, -18.85]$).

Furthermore, non-punishers returned a significantly higher proportion in the Trust Game than non-helpers did ($t(1007) = -9.43$, $p < 0.001$, $b = -26.82$, 95% CI = $[-32.40, -21.24]$), and non-actors who checked the cost returned significantly less than those who made an uncalculated decision not to help/punish ($t(1007) = -2.54$, $p = 0.01$, $b = -4.00$, 95% CI = $[-7.09, -0.91]$). There was also a significant interaction between deliberation and experiment ($t(1007) = 2.67$, $p = 0.008$, $b = 9.60$, 95% CI = $[2.53,

16.66], BF = 3.45), with uncalculated non-helpers returning the least (M = 15.69%, SD = 22.88), followed by calculated non-helpers (M = 21.29%, SD = 22.95), calculated non-punishers (M = 38.51%, SD = 22.42), and uncalculated non-punishers (M = 42.51%, SD = 21.65). However, when only participants with excellent comprehension were included, there no longer was a significant interaction (t(486) = 1.51, p = 0.13, b = 7.89, 95% CI = [−2.40, 18.17], BF = 0.46) although the average percentages returned remained similar (uncalculated non-helpers: (M = 13.74%, SD = 25.00), calculated non-helpers: (M = 17.14%, SD = 20.61), calculated non-punishers: (M = 36.27%, SD = 21.50), uncalculated non-punishers: (M = 40.76%, SD = 20.06)).

## Discussion

Previous work[26] has shown that helping behaviour that is performed in a reflexive or uncalculating manner can yield reputation benefits since observers infer that these actions reflect genuinely prosocial motives, rather than stemming from the rational calculation of costs and benefits. Accordingly, uncalculated help signals trustworthiness and people are more likely to behave in an uncalculated way when they are observed[26]. Over five experiments, we replicate this study and extend it by examining whether uncalculated punishment also leads to reputation improvements. In a further extension of previous work, we also ask whether punishers and helpers deliberate over the *impact to the target* (rather than the personal cost to themselves) and how such 'impact deliberation' is viewed by bystanders. In Study 1 (personal cost deliberation) we expected both uncalculated help and punishment to signal trustworthiness. In Study 2 (target impact deliberation) we expected uncalculated help to signal trustworthiness. Conversely, we expected *calculated* punishment to signal trustworthiness. As punishment inflicts harm on another, we expected that people would observe an implicit moral directive to deliberate over the harm they could inflict on another individual – and that individuals who inflict harm reflexively would be viewed negatively. Replicating previous results[26], we found that uncalculated help signals trustworthiness: helpers who did not consider the personal cost of helping were both more trusted and trustworthy than helpers who deliberated over the cost. Our punishment results were more mixed. Although punishers were more likely to perform uncalculated actions when observed, uncalculated punishment was not reliably associated with either perception of trustworthiness or with trustworthiness itself. Only uncalculating *non*-punishers were more trustworthy than calculating non-punishers. In contrast to the cost-checking context, considering the impact of helping had a larger impact on the trust and trustworthiness of non-helpers than helpers. Lastly, we found no conclusive evidence to suggest that checking the impact of punishing influences perceived or actual trustworthiness.

In Experiment 1, we replicated Jordan et al.'s[26], finding that uncalculating helpers were perceived as significantly more trustworthy than calculating helpers. Uncalculated helping provides a reliable signal of trustworthiness as it indicates that people are not considering the personal costs of helping and that helping stems from other-regarding rather than strategic motives. As in Jordan et al.[26], people were sensitive to these reputation benefits and were less likely to check the personal cost of helping when their decision process was observed than when it was hidden (H1.1). Finally, as in Jordan et al.[26], these reputation benefits were restricted to those who helped: deliberation had no effect on trust (H7.1) or trustworthiness (H15.1) when participants decided *not* to help.

We similarly expected uncalculated punishers to be perceived as, and to actually be, more trustworthy than those who deliberated over the personal cost of punishing (Exp. 2 & 3). We also expected people to be sensitive to these reputation consequences and to be less likely to check the personal cost (or to decide more quickly) when observed. These predictions were only partially supported. Participants were half as likely to check the cost of punishing when their decision process was observed (H1.2a) and were also significantly faster in their decision-making (H1.2b). In contrast to predictions, observers directionally trusted *calculating* punishers more than uncalculating punishers (while the Bayes Factor value for H2.2a indicated

support for this effect, frequentist statistics - which were preregistered as the primary decision criterion - did not support H2.2a or H2.2b). Trustworthiness results were also mixed. Uncalculating punishers were directionally more trustworthy than calculating punishers in Experiment 2 (H14.2a), but directionally less trustworthy in Experiment 3 (H14.2b). Note that Bayes Factor values indicate support for the null hypotheses, although power requirements were not met for H14.2a and H14.2b.

Whilst we expected both uncalculated help and uncalculated punishment to signal trustworthiness, we had diverging predictions around decisions *not* to act. Decision conflict over whether to help/punish could stem from self-interested considerations of whether to pay a cost. But unlike helping, punishment decision conflict could also stem from concerns about inflicting harm on the target. As participants initially believed punishing could potentially be free, we expected uncalculating non-punishers to be perceived as harm averse. Conversely, calculating decisions not to punish would indicate a selfish decision (the personal cost of punishing being too high). Support for these predictions was mixed. As expected, uncalculating non-punishers were more trustworthy than calculating non-punishers (though effects were only significant for Exp. 2, H15.2a and not Exp. 3, H15.2b). Perceived trustworthiness was not reliably affected, as observers directionally trusted uncalculating non-punishers more in Experiment 2 (H7.2a; the Bayes Factor value indicates null findings are inconclusive) but directionally trusted calculating non-punishers more in Experiment 3 (H7.2b; the Bayes Factor value indicates support for the null hypothesis).

Uncalculated punishment does not therefore seem to be perceived as a signal of trustworthiness – and uncalculated punishers were not more trustworthy. As deliberative decisions are often considered to be wise[48–50], uncalculated punishment might conceivably signal reduced competence[51], which could have affected perceived trustworthiness. While possible, this explanation is unlikely as the same ought to have been true for the helping context in Experiment 1. Alternatively, it is possible that the signalling effect of uncalculating punishment was too small to have been captured by the present work. However, several of the Bayes Factor values for null results in Experiments 2 and 3 were less than 0.33, supporting the absence of an effect as opposed to a need for more data. Moreover, we frequently found the directional opposite of our predictions, especially when deliberation was operationalised as decision time.

Helping may enhance reputation more than punishment because, even though third-party punishment is often viewed as a morally justified form of harm, people may still be unsure about those who engage in it[21,23,31]. Observers may therefore be unsure whether to trust punishers over non-punishers in the first place. Prior research has found that non-punishers can sometimes be trusted as much as punishers[24,30], and occasionally third-party punishers are even trusted less than non-punishers[32,52–55]. We found no significant difference in the perceived trustworthiness of punishers and non-punishers. Nevertheless, trustworthiness did vary. When restricting our sample to participants with excellent comprehension, punishers returned significantly more than non-punishers.

Perhaps punishment needs to be seen as the 'right thing to do' for the decision process to matter as a signal. This can be difficult, as punishment – unlike helping – is morally bad when undeserved, and there are also questions around legitimacy in that a fellow participant in an economic game may not be seen as an appropriate person to intervene[54,56]. Further, it has been argued that defection in economic games can be considered 'fair game', making the punishment of it less justified[31,57]. This is additionally important because the appropriateness of non-action decreases for more serious infractions[58]. Furthermore, even third-party punishers can be perceived as spiteful or competitive rather than prosocial, particularly when punishment is excessive[31]. However, punishment in this study is unlikely to be seen as excessive: punishing still leaves the defector with £0.15, the amount that they would have received had they themselves acted fairly. It is also unlikely that punishers are perceived as being competitive (aiming to increase their own payoffs relative to the defector's[5,59,60]) because, whether participants choose to punish or not, they always end with only one-third of the amount the defector receives (when participants punish, they finish the game with £0.05, whilst the defector

finishes with £0.15, and when participants do not punish, they finish with £0.10, whilst the defector finishes with £0.30).

It should be noted, that although helpers were indeed trusted significantly more than punishers (H4), they were not actually more trustworthy than punishers (H18). Our additional unregistered analyses further showed that non-punishers were significantly more trustworthy than non-helpers, which is in line with prior research[24]. Just as help is a more reliable signal of trustworthiness than punishment, not helping is a more reliable signal of *un*trustworthiness[21].

We expected deliberating about the *impact* of help/punishment to reflect negatively on helpers but positively on punishers, on the basis that considering both the prosocial aspect of third-party punishment and the negative consequences to the defector may be perceived as the 'right' thing to do when making punishing decisions. Helping on the other hand should be perceived as the 'right' thing to do, whether it helps a little or a lot. Helping without checking the impact on targets was therefore expected to signal trustworthiness. Conversely, deciding not to help without this consideration was predicted to result in particularly negative evaluations.

As predicted, uncalculating helpers were both more trusted (H6.1) and trustworthy (H19.1) than calculating helpers – but only directionally and with inconclusive Bayes Factors. Further, non-helpers who made calculating decisions were indeed more trustworthy than uncalculating non-helpers, but only significantly so when all participants were included (note: power requirements were not met for H20.1). Non-helpers were also directionally more trusted when they made calculating compared to uncalculating decisions (H11.1; the effect was statistically non-significant (the primary decision criterion) but supported by the Bayes Factor value). Counter to predictions, impact checking had a larger effect on the trust (H12.1) and trustworthiness (H13.1) of non-helpers compared to helpers. We do not find this result especially surprising. Indeed, we also argued that helping, regardless of the impact, is what matters the most for perceptions of (and actual) trustworthiness. It is possible that impact consideration matters more for those who decided *not* to help, as those who consider the impact at least considered helping, whereas those who do not consider the impact and do not help may be perceived as unwilling to help no matter how much another may benefit from it.

As predicted, considerations of target impact generally produced the opposite pattern in the punishing context. Counter to predictions, participants with excellent comprehension were significantly *less* likely to check the impact of punishing when they were observed than when their decision process was hidden (H5.2). Directionally, punishers who checked the impact to the target were perceived as (H6.2) and directionally were (H19.2) more trustworthy than uncalculating punishers (though effects were statistically non-significant). No conclusive results were found for non-punishers (H11.2 & H20.2). Bayes Factor values of non-significant analyses mostly indicated support for the null hypotheses (except H11.2 and H12.2, which were inconclusive, and H6.2 which indicates support for the alternative hypothesis).

## Limitations

The findings presented here should be interpreted within the context of certain limitations, particularly regarding ecological validity. The experimental design was highly abstract and therefore may not have fully captured the complexities of real-world decision-making processes related to trust and trustworthiness. Future studies could enhance ecological validity by employing scenarios or tasks more directly applicable to everyday situations, thereby potentially yielding more representative results. Additionally, past research emphasizes the importance of motive attributions in shaping evaluations of helpers and punishers[21,26,33], which the present study did not explicitly explore. Understanding the motives observers attribute to actors, as well as eliciting self-reported motives behind actors' calculated versus uncalculated decisions, could provide further insight into the mechanisms underlying third-party punishment and the extent to which it can be interpreted as a prosocial act. This could also help to differentiate between punishment and helping as signals of trustworthiness.

Lastly, there may be some concerns inherent to the Trust Game itself. We used this game to measure attitudes towards punishers as previous work has shown that punishment increases trustworthiness whereas results on whether punishers are 'liked' or rewarded for their actions are more mixed[16,23,28,61]. Nevertheless, it is important to acknowledge that the Trust Game does not fully disambiguate between trust and other underlying mechanisms. While decisions in the Trust Game can be partly attributable to risk attitudes[62], decision patterns in Trust and Risk games differ[63]. Differences in responses to help and punishment in the present study also indicate that observers did not make decisions based purely on risk preferences (if risk preferences were the key driver behind Trust Game decisions, we would not expect to observe any differences in help/punishment conditions).

## Data availability

All study data and materials, as well as the laboratory log are available on OSF under this link: https://osf.io/y2hgu/.

## Code availability

The analysis code is available on OSF under this link: https://osf.io/y2hgu/.

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

## Acknowledgements

This work was funded by a Royal Society University Research Fellowship and a Leverhulme Prize to NR, as well as the Leverhulme Trust as part of the Doctoral Training Program for the Ecological Study of the Brain (Grant No.

DS-2017-026) to NE. The funders had no role in study design, data collection and analysis, decision to publish or preparation of the manuscript.

## Author contributions

N.R. and N.E. developed the study concept. N.E. designed the study with revisions from N.R. and conducted data collection and analysis; N.E. drafted the initial manuscript; N.E. and N.R. revised and reviewed the manuscript and approved the final manuscript for submission.

## Competing interests

The authors declare no competing interest.

## Ethics

The research complies with all relevant ethical regulations. The study was approved by the UCL Ethics Board (Project ID: ICN-NH-PWB-7-1-23A). Informed consent was obtained from all participants. Although 'Player 3'/'The Sender' (Player A) and 'The Receiver' (Player B) really did exist and participants' decisions really did influence their own payoff and that of fellow participants, 'Player 1' (the cheated in punishing contexts or the recipient in helping contexts) and 'Player 2' (the violator) did not actually exist. Therefore, participants were fully debriefed after the study, and only those who previously indicated they were willing to take part in studies involving deception were invited to the study. Participants were compensated at an hourly rate of £9.
