## [Peer Review File · Communications Psychology]

20th Oct 23

Dear Nicole,

Thank you once again for your manuscript, entitled "Deliberating Cost and Impact: Trustworthiness Signals in Punishment and Helping". We are grateful for your patience and cooperation, especially as the evaluation of your Registered Report was affected by a change in requirements for the presentation of your work driven by changes in journal guidelines.

Before we issue Stage-1 in-principle acceptance, we ask you to undertake a final set of textual changes, which will ensure that the features of the Registered Report format are used most beneficially and that the completed Stage 2 Registered Report will be easy to read.

As you know, the Introduction of a Registered Report may not be altered following Stage 1 acceptance in principle (with the exception of changes in tense). It thus takes on a very special role in that it offers ultimate transparency with regard to the authors' hypotheses. The same goes for the analysis plan that is now included in the Stage 1 document and will form part of the Stage 2 Methods.

As mentioned in our correspondence last week, we have now implemented new guidelines on the inclusion of preregistered exploratory analyses. These are hypotheses that are ancillary to the main research question, for which it is possible but not guaranteed that the ultimate sample will be sufficiently powered at $>.95$. The new guidelines allow authors to feature these hypotheses and corresponding analyses as part of a Stage 1 Registered Report and include these in the Design table.

You currently include the ancillary preregistered exploratory hypotheses in the Supplementary Material. Once your Stage 2 Registered Report is complete, this division of information will be confusing to readers, and the location of the text means that it does not benefit from the same transparency with regard to preregistration as the Introduction. We therefore ask you to slightly condense the description of these hypotheses and move the text to the Introduction section.

We also ask you to mention how these hypotheses will be tested in the Analysis plan under the heading Preregistered Exploratory Analyses. You must clearly state in the Analysis plan that power of $>.95$ for these hypotheses cannot be guaranteed a priori and that if power requirements are not met, the results will be reported as suggestive, pending confirmation in future research. In your Stage 2 Registered Report, the results of these analyses should be listed under the heading "Preregistered Exploratory Analyses".

Finally, there may be only one Design Table, which needs to be Table 1. You must remove the copy of the Design Table from Supplementary Information file. Please ensure that for each preregistered exploratory hypothesis, you include a statement in the interpretation column of the Design Table that if power requirements are not achieved, the results will be reported as suggestive, pending confirmation in future research. To ensure that the Design Table can be listed as Table 1 in the final paper, please change the order of mention of the Tables in the main text.

* In your cover letter, please include the following information:

- An anticipated timeline for completing the study if your Stage 1 submission is accepted in principle.
- A statement confirming that you agree to share your raw data, any digital study materials, computer code, and laboratory log for all eventually published results.
- A statement confirming that, following Stage 1 in principle acceptance, you agree to register your approved protocol on the Open Science Framework (<https://osf.io/>) or other recognised repository, either publicly or under private embargo, until submission of the Stage 2 manuscript.
- A statement confirming that if you later withdraw your paper, you agree to the Journal publishing a short summary of the pre-registered study under a section Withdrawn Registrations.

[link redacted]

We hope to receive your revised manuscript within two weeks. If you cannot send it within this time, please let us know.

* **TRANSPARENT PEER REVIEW:** Communications Psychology uses a transparent peer review system. This means that we publish the editorial decision letters including Reviewers' comments to the authors and the author rebuttal letters online as a supplementary peer review file. We publish these records for all accepted manuscripts. However, on author request, confidential information and data can be removed from the published reviewer reports and rebuttal letters prior to publication. If your manuscript has been previously reviewed at another journal, those Reviewers' comments would not form part of the published peer review file.

Communications Psychology is committed to improving transparency in authorship. As part of our efforts in this direction, we are now requesting that all authors identified as 'corresponding author' on published papers create and link their Open Researcher and Contributor Identifier (ORCID) with their account on the Manuscript Tracking System (MTS), prior to acceptance. ORCID helps the scientific community achieve unambiguous attribution of all scholarly contributions. You can create and link your ORCID from the home page of the MTS by clicking on 'Modify my Springer Nature account'. For more information please visit www.springernature.com/orcid.

We look forward to seeing the revised manuscript.

Sincerely,
Marike

Marike Schiffer, PhD
Chief Editor
Communications Psychology

13th Nov 23

Dear Nicole,

Thank you once again for submitting your revised Stage 1 Registered Report, entitled "Deliberating Cost and Impact: Trustworthiness Signals in Punishment and Helping." Everything is in order and I am delighted to say that we can offer acceptance in principle. You may progress to Stage 2 and complete the study as approved.

As you know, a condition of in-principle-acceptance is that the authors agree to deposit their Stage 1 accepted protocol in a repository, either publicly or under embargo until Stage 2 acceptance and publication. We are very keen to showcase our in-principle accepted protocols, so that our readers, reviewers, and potential authors can gain insight into the requirements of the format as well as an idea of the types of projects that are suitable for publication in Communications Psychology. We have set up a space on figshare to host all of our in-principle accepted protocols, which can either be made public or kept under embargo until Stage 2 acceptance (depending on author preference). This gives you the opportunity to have your work publicly associated with Communications Psychology, and of course we will be very pleased to showcase your report if you agree to share it publicly.

Depositing the work on our figshare space does not preclude deposition of your Stage 1 protocol on other depositories – your protocol can also be posted on OSF, Dataverse, Dryad or any other public repository of your choice. You also do not need to do anything – if you agree with posting your protocol on our figshare space, we will upload your protocol on your behalf and either set it public or place it under embargo, depending on your choice. Your protocol will be licensed under a CC BY license (Creative Commons Attribution 4.0 International License). The CC BY license allows for maximum dissemination and re-use of open access materials and is preferred by many research funding bodies. Under this license users are free to share (copy, distribute and transmit) and remix (adapt) the contribution including for commercial purposes, providing they attribute the contribution in the manner specified by the author or licensor (read full legal code: <http://creativecommons.org/licenses/by/4.0/legalcode>) Please note that any use of <https://springernature.figshare.com> will be subject to the Figshare terms of use. Figshare has the right to enforce these terms and conditions where applicable. Use of third party services and sites will be subject to the relevant terms of use and will apply if we act on your behalf in this regard. Do let me know if you would like to take up this option or if you have any questions regarding the protocol deposition requirement.

IMPORTANT:

In cases where the registered experimental design is altered after AIP due to unforeseen circumstances (e.g. change of equipment or unanticipated technical error), the authors should consult the editors immediately for advice, prior to the completion of data collection.

Following completion of your study, we invite you to resubmit your paper for peer review as a Stage 2 Registered Report. Please note that your manuscript can still be rejected for publication at Stage 2 if the Editors consider any of the following to hold:

- The results were unable to test the authors' proposed hypotheses by failing to meet the approved outcome-neutral criteria

- The authors altered the Introduction, rationale, or hypotheses, as approved in the Stage 1 submission
- The authors failed to adhere closely to the registered experimental procedures without previously seeking editorial approval
- Any post hoc (unregistered) analyses were either unjustified, insufficiently caveated, or overly dominant in shaping the authors' conclusions
- The authors' conclusions were not justified given the data obtained

We encourage you to read the complete guidelines for authors concerning Stage 2 submissions at <https://www.nature.com/commspsychol/submit/registered-reports> and <https://www.nature.com/documents/commspsychol-style-formatting-checklist-article-rr.pdf>.

Please especially note the requirements for protocol deposition, data sharing, and that withdrawing your manuscript will result in publication of a Retracted Registration.

When you are ready, please use the following link to access your home page and submit your Stage 2 Registered Report:

[link redacted]

*This url links to your confidential homepage and associated information about manuscripts you may have submitted or be reviewing for us. If you wish to forward this e-mail to co-authors, please delete this link to your homepage first.

* **TRANSPARENT PEER REVIEW:** Communications Psychology uses a transparent peer review system. This means that we publish the editorial decision letters including Reviewers' comments to the authors and the author rebuttal letters online as a supplementary peer review file. This means that the records will be published together with your Stage 2 report. On author request, confidential information and data can be removed from the published reviewer reports and rebuttal letters prior to publication. If your manuscript has been previously reviewed at another journal, those Reviewers' comments would not form part of the published peer review file.

We expect your Stage 2 Registered Report to be submitted by the date specified in your latest cover letter. If unforeseen circumstances prevent submission by that date, please contact us as soon as possible to discuss any changes to the submission time-frame.

Thank you again for offering us this work and we look forward to receiving your Stage 2 Registered Report.

Best wishes
Marike

Marike Schiffer, PhD
Chief Editor
Communications Psychology

13th Mar 24

Dear Nicole,

Thank you once again for submitting your Stage 2 Registered Report, entitled "Deliberating Cost and Impact: Trustworthiness Signals in Punishment and Helping," and for your patience during the re-review process.

Your manuscript has now been evaluated by Reviewers 1 and 3 from the previous rounds of review, whose comments are included at the end of this letter. In the light of our reviewers' advice, we are pleased to inform you that we will be able to accept your Stage 2 manuscript, pending revisions to address Reviewers' comments and editorial requests.

One of the main reasons for delays in eventual acceptance is failure to fully comply with editorial policies and formatting requirements. To assist you with finalizing your manuscript for publication, I attach our Editorial Requests Table which lists all of our editorial policies and formatting requirements.

Regarding Reviewer #1's feedback, please ensure that the interpretation of the results, including the criterion by which a finding is deemed credible evidence or not, must follow the preregistered commitment to data interpretation. Second, please ensure that your Discussion includes the obligatory "Limitations" section and use this section to refer back to potential alternative explanations of behaviour in the task. Finally, in the context of a Registered Report it makes sense to list participant details in the Results, but the journal guidelines are not prescriptive on this issue either for Articles or Registered Reports.

Please attend to *every item* in the Table and upload a copy of the completed checklist with your submission. I have highlighted in the checklist items that require your attention. I also mention here a few points that are frequently missed and can cause delays:

1) Ensure that all corresponding authors have linked their ORCID to their account on our online manuscript handling system. This is very frequently missed and invariably causes delays in formal acceptance.

2) Ensure that you provide all of the materials requested in the attached checklist and below with your final submission.

OPEN ACCESS:

Communications Psychology is a fully open access journal. Articles are made freely accessible on publication under a CC BY license (Creative Commons Attribution 4.0 International License). This license allows maximum dissemination and re-use of open access materials and is preferred by many research funding bodies.

For further information about article processing charges, open access funding, and advice and

support from Nature Research, please visit <https://www.nature.com/commspsychol/article-processing-charges>

At acceptance, you will be provided with instructions for completing this CC BY license on behalf of all authors. This grants us the necessary permissions to publish your paper. Additionally, you will be asked to declare that all required third party permissions have been obtained, and to provide billing information in order to pay the article-processing charge (APC).

* **CODE AVAILABILITY:** To proceed to formal acceptance, you must now publicly deposit the custom analysis code supporting your conclusions; please use a repository that mints the code with a digital object identifier (DOI). The manuscript must include a section titled "Code Availability" at the end of the methods section. The link to the repository and the DOI must be included in the Code Availability statement. Publication as Supplementary Information will not suffice.

* **DATA AVAILABILITY:**

It is a requirement for Registered Reports to make data publicly available. All Communications Psychology manuscripts must include a section titled "Data Availability" at the end of the Methods section. More information on this policy, is available in the Editorial Requests Table and at <http://www.nature.com/authors/policies/data/data-availability-statements-data-citations.pdf>. Please share a link to your publicly deposited data in the Data Availability statement.

We hope to hear from you within [ENTER TIME PERIOD]; please let us know if the revision process is likely to take longer.

Please use the following link for uploading the materials:

[link redacted]

With best regards,

Marika Schiffer, PhD
Chief Editor
Communications Psychology

Reviewer #1:

Remarks to the Author:

I previously reviewed this Registered Report for Nature Human Behaviour. In my previous reviews, I raised some concerns (e.g., about construct validity and power considerations), but overall had a

positive impression. My impression about the Stage 2 Registered Report is similar.

I've organized my feedback following the evaluation categories provided by the editor.

#####

Whether the data are able to test the authors' proposed hypotheses by passing the approved outcome-neutral criteria (such as absence of floor and ceiling effects or success of positive controls)

#####

This is an issue that I feel should only be evaluated at Stage 1. At that time, I outlined potential alternative explanations in regards to the presumed underlying mechanism – but I acknowledged that the author's assumptions nonetheless seemed reasonable.

I don't believe that Nature Communications Psychology asked the authors to incorporate that feedback at Stage 1 (which is fine). So, I do not think it is fair to bring it up again at Stage 2, which should be focused on the extent to which authors followed the methodological plan.

#####

Whether the Introduction, rationale and stated hypotheses are the same as the approved Stage 1 submission

#####

The authors made mostly minor changes to the Stage 1 text (e.g., updating the tense used when describing the methodology). There are, however, two changes worth flagging.

First, the study numbers for two of the hypotheses were changed. I'm guessing that the authors were correcting a previous typo, but it would be helpful to have some clarification.

Second, participant demographics were added to the Sampling Plan section. I do not personally think this is an issue. However, some Registered Report editors have argued that (a) little-to-no changes should be made to this section at Stage 2, and (b) that this information should instead be moved to the Results section.

#####

Whether the authors adhered to the registered experimental procedures

#####

There was nothing in the paper that led me to believe that they deviated from their pre-registered experimental procedures.

However, the authors did not consistently follow their pre-registered inference criteria. The authors planned to use Bayesian analyses to characterize the strength of evidence, but most of the inferences only discuss the presence (or lack thereof) of an effect. This is particularly important for a few cases where the frequentist and Bayesian analyses lead to conflicting inferences. For example, the Experiment 3 test of H1.2b yielded a p-value of .007 (which the pre-registered analysis plan said would be interpreted as supportive evidence) and a Bayes Factor of 2.16 (which the pre-registered analysis plan said would be interpreted as being inconclusive).

#####

Whether any unregistered exploratory analyses added by the authors are justified, methodologically sound, and informative

#####

The exploratory analyses seemed reasonable to me.

#####

Whether the authors' conclusions are justified given the data

#####

Given that it was part of their pre-registered plans, I think the authors are obligated to update their conclusions to reflect both the planned frequentist and Bayesian analyses. They often did so, but there are a few cases (e.g., the ones I described above) where they did not.

This is a minor point, of course.

#####

Concluding thought

#####

I can tell a ton of work went into this project. Congrats to the authors for nearing the finish line.

Reviewer #3:

Remarks to the Author:

I don't have any further comments on this manuscript beyond what I gave at the earlier stage.

It reads well and seems thorough and sound.

Response to Reviewers

We are very grateful to the reviewers for their renewed careful reading of our manuscript and their comments. We have carefully revised the manuscript according to the comments we received.

Point-by-point replies to reviewer comments are provided below. To clearly differentiate between reviewer comments and author responses, the former will be italicised.

Many thanks,
Nicole Engeler and Nichola Raihani

Reviewer #1:

Remarks to the Author:

I previously reviewed this Registered Report for Nature Human Behaviour. In my previous reviews, I raised some concerns (e.g., about construct validity and power considerations), but overall had a positive impression. My impression about the Stage 2 Registered Report is similar. I've organized my feedback following the evaluation categories provided by the editor.

Thank you for this evaluation. We believe that previous concerns were addressed. For instance, when concern was expressed that decisions not to punish others could stem either from harm aversion or from an aversion to paying costs, we changed the minimum cost of punishing/helping to be £0.00 rather than £0.01, to disambiguate aversion to paying personal costs from harm aversion. Based on concerns that our study could not address nuanced reactions people have to uncalculated and calculated punishment versus helping, we added helping conditions to both studies. To manage power considerations we agreed to label hypotheses as exploratory whenever power might not be met – which we made sure to do throughout.

#####

Whether the data are able to test the authors' proposed hypotheses by passing the approved outcome-neutral criteria (such as absence of floor and ceiling effects or success of positive controls)

#####

This is an issue that I feel should only be evaluated at Stage 1. At that time, I outlined potential alternative explanations in regards to the presumed underlying mechanism – but I acknowledged that the author's assumptions nonetheless seemed reasonable.

I don't believe that Nature Communications Psychology asked the authors to incorporate that feedback at Stage 1 (which is fine). So, I do not think it is fair to bring it up again at Stage 2, which should be focused on the extent to which authors followed the methodological plan.

Thank you for this assessment.

#####

Whether the Introduction, rationale and stated hypotheses are the same as the approved Stage 1 submission

#####

The authors made mostly minor changes to the Stage 1 text (e.g., updating the tense used when describing the methodology). There are, however, two changes worth flagging.

First, the study numbers for two of the hypotheses were changed. I'm guessing that the authors were correcting a previous typo, but it would be helpful to have some clarification.

Second, participant demographics were added to the Sampling Plan section. I do not personally think this is an issue. However, some Registered Report editors have argued that (a) little-to-no changes should be made to this section at Stage 2, and (b) that this information should instead be moved to the Results section.

Thank you for pointing this out. The changes (besides the required updates) were indeed corrections of typos. We have also moved the demographics section to the beginning of the results section so that the Stage 1 protocol remains as unchanged as possible.

#####

Whether the authors adhered to the registered experimental procedures

#####

There was nothing in the paper that led me to believe that they deviated from their pre-registered experimental procedures.

However, the authors did not consistently follow their pre-registered inference criteria. The authors planned to use Bayesian analyses to characterize the strength of evidence, but most of the inferences only discuss the presence (or lack thereof) of an effect. This is particularly important for a few cases where the frequentist and Bayesian analyses lead to conflicting inferences. For example, the Experiment 3 test of H1.2b yielded a p-value of .007 (which the pre-registered analysis plan said would be interpreted as supportive evidence) and a Bayes Factor of 2.16 (which the pre-registered analysis plan said would be interpreted as being inconclusive).

Thank you for raising this important point. We have now made sure to include more comments on Bayesian results, and especially highlighted instances in which frequentist and Bayesian analyses conflicted with each other.

#####

Whether any unregistered exploratory analyses added by the authors are justified, methodologically sound, and informative

#####

The exploratory analyses seemed reasonable to me.

Thank you for this assessment.

#####

Whether the authors' conclusions are justified given the data

#####

Given that it was part of their pre-registered plans, I think the authors are obligated to update their conclusions to reflect both the planned frequentist and Bayesian analyses. They often did so, but there are a few cases (e.g., the ones I described above) where they did not. This is a minor point, of course.

Thank you for your careful considerations. We amended our conclusions as mentioned above.

#####

Concluding thought

#####

I can tell a ton of work went into this project. Congrats to the authors for nearing the finish line.

Thank you very much for this kind statement!

Reviewer #3:

Remarks to the Author:

I don't have any further comments on this manuscript beyond what I gave at the earlier stage.

It reads well and seems thorough and sound.

Thank you very much for these positive comments! To address comments made at an earlier stage, we included concerns around the Trust Game in the Limitation section.